# The Genetic Control of SEEDSTICK and LEUNIG-HOMOLOG in Seed and Fruit Development: New Insights into Cell Wall Control

**DOI:** 10.3390/plants11223146

**Published:** 2022-11-17

**Authors:** Maurizio Di Marzo, Nicola Babolin, Vívian Ebeling Viana, Antonio Costa de Oliveira, Bruno Gugi, Elisabetta Caporali, Humberto Herrera-Ubaldo, Eduardo Martínez-Estrada, Azeddine Driouich, Stefan de Folter, Lucia Colombo, Ignacio Ezquer

**Affiliations:** 1Dipartimento di Bioscienze, Università degli Studi di Milano, Via Celoria 26, 20133 Milano, Italy; 2Plant Genomics and Breeding Center, Federal University of Pelotas, Capão do Leão 96010-610, RS, Brazil; 3Laboratoire Glycobiologie et Matrice Extracellulaire Végétale EA4358, UNIROUEN—Universitè de Rouen Normandie, 76000 Rouen, France; 4Unidad de Genómica Avanzada (UGA-LANGEBIO), Centro de Investigación y de Estudios Avanzados del Instituto Politécnico Nacional (CINVESTAV-IPN), Km. 9.6 Libramiento Norte, Carretera Irapuato-León, Irapuato 36824, Guanajuato, Mexico; 5Fédération de Recherche “NORVEGE”-FED 4277, 76000 Rouen, France

**Keywords:** cell wall, transcriptional regulators, pectins, seed growth, fruit, germination

## Abstract

Although much is known about seed and fruit development at the molecular level, many gaps remain in our understanding of how cell wall modifications can impact developmental processes in plants, as well as how biomechanical alterations influence seed and fruit growth. Mutants of *Arabidopsis thaliana* constitute an excellent tool to study the function of gene families devoted to cell wall biogenesis. We have characterized a collection of lines carrying mutations in representative cell wall-related genes for seed and fruit size developmental defects, as well as altered germination rates. We have linked these studies to cell wall composition and structure. Interestingly, we have found that disruption of genes involved in pectin maturation and hemicellulose deposition strongly influence germination dynamics. Finally, we focused on two transcriptional regulators, SEEDSTICK (STK) and LEUNIG-HOMOLOG (LUH), which positively regulate seed growth. Herein, we demonstrate that these factors regulate specific aspects of cell wall properties such as pectin distribution. We propose a model wherein changes in seed coat structure due to alterations in the xyloglucan-cellulose matrix deposition and pectin maturation are critical for organ growth and germination. The results demonstrate the importance of cell wall properties and remodeling of polysaccharides as major factors responsible for seed development.

## 1. Introduction

Seeds are of critical importance to ecology and agronomy, and constitute an efficient mechanism for genetic transmission through generations [1]. Seed production is a critical factor in agriculture. The number of seeds in a fruit and the seed quality (in terms of mass, size, and germinability) have been the object of breeding programs for many years. The molecular pathways controlling both seed yield and seed quality have been the focus of a number of studies [2,3]. Angiosperm seeds originate as a result of sexual reproduction [4,5,6]. In seeds of flowering plants, three major compartments can be distinguished: embryo, endosperm, and seed coat [2,7,8]. The embryo is a potential plantlet and is surrounded by a nutritive tissue called endosperm [9]. Embryo and endosperm are derived from individual fertilization events (a distinctive feature of angiosperms) and develop while embedded in maternal tissues that form the seed coat, an outer protective layer formed by dead tissue [10]. This structure has been shown to be critical for the entire seed life cycle, including seed growth and germination. The final seed size is determined by the extent of endosperm development, growth of the embryo, and differentiation of the seed coat, all of which must be coordinated strictly to regulate seed germination [11]. In order to germinate properly, the first requirement for a seed is to recover from a desiccated state. Following that, metabolic processes are restarted, activating cellular events that lead to embryo growth, testa rupture, and radicle emergence [1,12]. In the last step, endosperm rupture and radicle protrusion occur through the seed coat, completing the germination process [13,14,15]. Mechanically, two distinct and opposing forces control seed germination: embryo growth and the strength of the seed coat. When the potential force of embryo growth exceeds the mechanical force of the seed coat by increasing extensibility of embryo cell wall (CW) and allowing a plastic extension (therefore inducing mechanical rupture of testa), germination is triggered [12,16].

CW-remodeling enzymes are involved in CW biogenesis, reinforcement and loosening [17]. The major CW glycan polymers are cellulose, hemicellulose, and pectins. Cellulose is synthesized by the cellulose synthase (CESA) complexes (CSCs) that are built in the Golgi apparatus and transported to the plasma membrane where they synthetized this polymer, while hemicellulose and pectins are synthesized in the Golgi apparatus [18,19,20,21]. Pectin and hemicellulose (e.g., xyloglucan) are transported and deposited into the CW via the SYNTAXIN OF PLANTS 61 trans-Golgi network compartment pathway [22]. Cellulose microfibrils are present in organized crystalline regions. Hemicellulose polymers form direct interconnections with cellulose microfibrils, reinforcing the CW [23]. Pectin, together with hemicelluloses, comprises the hydrated matrix localized between cellulose microfibril spaces. The pectic matrix is formed by four main galacturonan structural domains: homogalacturonan (HG), xylogalacturonan (XGA), rhamnogalacturonan I (RG-I), and rhamnogalacturonan II (RG-II) [24]. Recent evidence suggests that pectins can also form direct interconnections with cellulose microfibrils [23,25].

The role of CW in seed size and germination control is quite well documented [8,26,27,28]. Different transcription factors (TFs) have been shown to influence seed development via CW structural modifications. For instance, using atomic force microscopy approaches together with molecular and CW chemical analyses, the role of SEEDSTICK (STK), a MADS box TF, in seed coat biophysical properties was recently described [29]. STK controls pectin methylesterase (PME) activity and pectin maturation, thus influencing seed germination [29]. STK is a positive regulator of seed growth, since a reduction in seed size has been reported in the *stk* mutant [30]. Other transcriptional regulators, such as GLABRA 2 (GL2), MYB DOMAIN PROTEIN 52 (MYB52), and LEUNIG HOMOLOG (LUH), also known as MUCILAGE MODIFIED1 (MUM1), also control PME activities [31,32,33]. *GL2* encodes a TF required for the proper differentiation of several epidermal cell types and regulates testa identity [34]. *LUH* is a major regulator of the seed coat development, and was reported to act as a transcriptional activator of genes required for mucilage extrusion *MUCILAGE MODIFIED 2* (*MUM2*), *SUBTILISIN PROTEASE1.7* (*SBT1.7*), and *BETA-XYLOSIDASE 1* (*BXL1*) [10,31,35]. Recently, the role of LUH as a transcriptional repressor of *Tubby-like Protein 2,* which regulates HG biosynthesis in seed coat mucilage, was also proposed [36].

Effective seed development relies on the complex interaction between TFs (and their downstream targets), hormonal regulators, and environmental factors [2,3,37]. The molecular network controlling epidermal CW development of the seed coat has recently been elucidated. Upstream players such as LUH, STK, and GL2 regulate the function of downstream targets. These code for proteins that affect CW structure, such as CESA5 and FEI2 (leucine-rich repeat kinase protein), which act in cellulose biosynthesis and deposition. PECTIN METHYLESTERASE INHIBITOR 6 (PMEI6), subtilisin-like serine protease SBT1.7, E3 ubiquitin ligase FLYING SAUCER 1 (FLY1), and BXL1 regulate pectin maturation and remodeling [23,38,39,40,41,42,43]. It was recently shown that the pectin methylesterification pattern controlled by PMEI6 is necessary for class III peroxidase PEROXIDASE 36 (PER36) stable anchoring to pectins. PER36 loosens the CW and is fundamental for proper mucilage release [44]. Another actor involved in CW rearrangement is *ALPHA XYLOSIDASE 1* (*XYL1*). *XYL1* encodes the only Arabidopsis α-xylosidase involved in the xyloglucan metabolism [45]. Recently, it was reported that the *XYL1* gene is directly targeted by STK, which regulates its expression, leading to proper seed and silique growth [28]. Over the last few years, several developmental studies have demonstrated that fruit developmental processes require major changes in plant CW composition [28,46,47,48,49]. Silique formation in Arabidopsis is a dynamic process during which both seed and silique development grow in concert in a tightly coordinated manner. Upon fertilization, carpel cells undergo division, expansion, and differentiation to develop the different tissues that form the mature siliques [50,51]. The final stages of silique maturation require CW structural modifications in order to establish a dehiscence zone, which enables seed dispersal [52,53]. However, the complex molecular network controlling CW remodeling, which affects seed and fruit growth, is still not fully understood, although recent work suggests that common regulators might coordinate both processes [28].

The main goal of this work is to understand how mutations of critical transcriptional regulators and CW-related genes could impact the developmental processes in the reproductive structure. We show that mutation of transcriptional regulators of the seed coat alters pectin structures and composition, and we demonstrate that CW processes are crucial for seed size, fruit growth, and germination.

## 2. Results

### 2.1. Cell Wall Structural Alterations Influence Seed and Silique Size

Seed size, together with ovule number and pistil/fruit growth, are critical components of total seed yield [37,54,55]. Developing tissues in plants contain a mechanically strong but extensile complex structure formed by CWs, which contain cellulose microfibrils as well as hemicellulosic and pectic polysaccharides [17,56]. In order to study the effect of disruption of CW structure on developing and growing tissues, we selected mutants that display mechanical disruptions of any of these elements, and we measured seed and silique size.

First, we tested mutants for genes involved in cellulose biosynthesis or deposition. The *fei2.1* mutant plants produced smaller seeds, considering the area size, due to a shorter length and width of the seed itself. This is also coupled with a smaller silique with respect to WT (Figure 1a–d). The *cesa5* cellulose synthase mutant displayed smaller seeds compared to WT, as previously observed [57], which was caused by a reduction in seed length (Figure 1a,b). However, *cesa5* showed a silique length similar to WT (Figure 1c,d). These results suggest that both *FEI2* and *CESA5* act as positive regulators of seed size, highlighting the importance of cellulose deposition in developing seeds. However, just *FEI2* could be considered a positive regulator of fruit growth.

We then checked developmental defects in mutants of genes involved in pectin maturation. Mutation in the pectin methylesterase inhibitor *PMEI6* (*pmei6*) resulted in production of smaller seeds with reduced length and width (Figure 1a,b). Nevertheless, silique length appears to be WT-like (Figure 1c,d). Since single PMEs and PMEIs in Arabidopsis development have highly specific spatiotemporal functions [29], we decided to test the developmental impact on seed and silique in two *fly1* (RING/U-box superfamily protein) mutant alleles. *FLY1* has been proposed to regulate the degree of methylesterification of pectin by recycling unprocessed PME enzymes in the endomembrane system of the seed coat epidermis [39]. Interestingly, we found that mutations in *fly1.1* and *fly1.3* both decreased seed area, but in two different manners. *fly1.1* seeds presented a reduction in width, while *fly1.3* seeds showed a reduction in the length compared to WT (Figure 1a,b). However, only the *fly1.3* mutation increased the silique length, while *fly1.1* displayed a WT-like silique (Figure 1c,d). Finally, mutations in class III peroxidase *PER36* gene (*per36*) lead to smaller seeds due to a reduction in length and width, while siliques showed a WT-like length phenotype (Figure 1a–d). Overall, it seems that pectin modifications (e.g., pectin methylesterification) can influence developmental processes in particular seed growth and, to a minor extent, growth of the fruit. In fact, the two tested *fly* mutant alleles were characterized by a reduced seed size phenotype; thus, *FLY1* acts as a positive regulator during seed development and acts as negative regulator in silique development, whilst *PER36* only positively influences seed growth.

We also intended to analyze the developmental defects caused by alterations in hemicellulose deposition. The main hemicellulose in most dicot species is xyloglucan [58]. Interestingly, we found that *bxl1* dramatically affected seed size. The *bxl1* mutation altered seed size, producing smaller seeds, with a significant reduction in width and a slight increase in length compared to WT (Ws Wassilewskija) (Figure 1a,b). The *bxl1* mutant did not show differences in silique length with respect to WT (Figure 1c,d). Overall, it appears that *BXL1* acts as a positive regulator of seed growth.

### 2.2. Transcriptional Cascades Controlling Cell Wall Modifications Are Important for Seed and Fruit Development

In order to further investigate CW effects on seed development, we analyzed critical transcriptional regulators controlling seed coat differentiation. As previously described, the *stk* mutant plants produced smaller siliques [28,30,59], as well as smaller seeds when compared to WT [28,30,60]. We decided to analyze the involvement of the transcriptional co-activator LUH on seed and fruit development. The *luh* mutant also displayed a reduction in seed area due to a smaller width, but, conversely to *stk*, silique length was similar to WT (Figure 1a–d). These data suggest that LUH acts as a positive developmental regulator of seed growth.

Mutation of *GL2* only caused effects on seed development. In fact, *gl2-8* mutant plants produced seeds with smaller lengths and widths, and, thus, with an overall reduced seed size with respect to the WT (Figure 1a,b). In contrast, *gl2-8* was able to produce normal siliques (Figure 1c,d). These findings suggested that GL2 is required for normal seed production and does not have an impact on silique growth. 

Overall, these data indicate that these regulatory factors control the size and shape of the seed by different mechanisms. 

### 2.3. Dual Disruption of STK and LUH Impacts Seed Development

The transcriptional regulators LUH and STK have been shown to regulate pectin demethylesterification in seed coat mucilage [10,29,31,61]. They control the pathway in different directions: *LUH* activates the transcription of the PME modifiers *PMEI6*, *SBT1.7*, and *FLY1* [10,31,61], while STK positively regulates the transcription of *PMEI6* and negatively regulates *SBT1.7* [29]. Recently, it was shown that STK and LUH repress each other [29]. 

Both *LUH* and *STK* mutations negatively influence seed size, albeit in different ways. Since mutation for *STK* influences the length [28], while mutation for *LUH* affects the width of the seeds (Figure 1a,b and Figure 2b), we decided to investigate the effects of double mutations for these two transcriptional regulators on seed growth. The *luh stk* seeds presented developmental defects, with the presence of seeds that displayed irregular shapes and some seeds presenting “wrinkled” phenotypes (63%). The rest (37%) presented a ”non-wrinkled” phenotype (Figure 2a and Appendix A). We performed a detailed analysis of the surface of seed epidermis in these genotypes (Figure 2a). WT, *stk,* and *luh* single mutants presented a typical regular (pentagonal or hexagonal) appearance, with thick radial cell walls and a central columella (Figure 2a). Observation of the “wrinkled” phenotype of *luh stk* seeds highlighted epidermal cells with a depression in the center of the columella (Figure 2a).

The *luh stk* double mutant seeds displayed an intermediate length phenotype between the *stk* and *luh* mutants (shorter than *luh*, longer than *stk*), but a clear increase in width when compared to the single mutants (Figure 2b,c). This allows *luh stk* to rescue the seed area defects observed in the single mutants. Overall, it seems that control of seed size downstream from these regulators follows independent pathways. On the other hand, *luh stk* displayed shorter siliques (as observed for the *stk* mutant without statistical differences), while *luh* developed WT-like siliques (Figure 2d,e). These phenotypes suggest that STK regulates fruit development independently from *LUH*, which does not influence fruit growth.

### 2.4. The stk and luh Mutant Seeds Have Altered Cell Wall Composition

In order to elucidate the impact of CW components on seed growth, we analyzed the monosaccharide composition of the CW extracted from mature seeds of *luh* and *stk* single mutants, as well as the *luh stk* double mutant, compared to WT (Figure 3). Compositional analysis showed that the mature seed fractions of both WT, *luh*, *stk,* and *luh stk* consisted predominantly of rhamnose (Rha), arabinose (Ara), and galacturonic acid (GalA), each representing approximately 20–25% of total monosaccharides (Figure 3a). While the levels of Ara and GalA were similar among the different genotypes, Rha levels were more variable. The *luh stk* double mutant contained reduced Rha with respect to the WT (20.3% vs. 24.0%), but not to the single mutants (Figure 3a).

Monosaccharides such as galactose (Gal) and xylose (Xyl) were also relatively abundant (Figure 3a). Xyl levels were statistically similar among the four genetic backgrounds tested (Figure 3a). Gal, together with Ara, are the most abundant monosaccharides present in the side chains of RG-I, a highly branched pectic polysaccharide (Figure 3a). Interestingly, we found that both single mutants *luh* and *stk* presented statistically higher levels of Gal (14.6 and 13.7%) with respect to the WT (12.7%, Figure 3a). The *luh stk* double mutant presented high Gal levels (17.1%) with respect to the other genotypes (Figure 3a), indicating a possible additive effect of STK and LUH disruption.

Of the remaining CW monosaccharides detected, Fucose (Fuc), Glucuronic Acid (GlcA), Mannose (Man), and Glucose (Glc) were present in minor fractions (Figure 3a). There were no differences between genotypes concerning Fuc or Man (Figure 3a). Notably, the *luh* mutant displayed higher levels of GlcA compared to WT (2.7% vs. 1.8%). The difference was attenuated upon *STK* disruption; *luh stk* contained 2% of GlcA (the single *stk* mutant had reduced GlcA (1.4%) relative to WT, although not statistically different) (Figure 3a). This indicated opposing effects of *STK* versus *LUH* disruption of GlcA composition. Measurements of Glc content showed that the *luh stk* double mutant presented statistically higher Glc levels compared to WT (2.6%), *luh* (3.2%), and *stk* (3%) (Figure 3a). In order to better understand possible differences in pectin composition, specific ratios were calculated based on the relative amounts of the main pectin components (Figure 3b). The ratio of GalA to Rha reflects the proportion between homogalacturonan (HG) and rhamnogalacturonan I (RG-I), as these two monosaccharides are unique monomers of their respective pectin backbones. We found that there were no statistical differences in the GalA/Rha ratio between any of the genetic backgrounds (Figure 3b). In addition, we assessed the degree of branching of RG-I based on both the ratio of Ara to Rha and the ratio of Gal to Rha (Gal and Ara being monosaccharides present in RG-I side chains and Rha being a central constituent of the backbone). *luh* CWs displayed a higher Gal/Rha ratio (28% increase compared to WT), while *stk* was not statistically significant (Figure 3b). We found that the *luh stk* double mutant had a marked and additive increase in the Gal/Rha ratio (59% increase with respect to WT). We also found that *luh* and *stk* CWs displayed a higher Ara/Rha ratio (18% and 9% increase, respectively); however, this was not statistically significant when compared to the WT (Figure 3b). The *luh stk* double mutant showed an increase in the Ara/Rha ratio (19% increase with respect to WT). 

### 2.5. LUH and STK, Contrasting Control of Pectin Composition and Ramification

The altered composition in monosaccharides described previously likely derives from alterations of the pectic backbone. In order to confirm this, we performed a detailed compositional analysis of exclusively pectin-enriched extracts from mature seeds (Figure 4a,b). 

The three main components of pectins in WT were, approximately, Rha (30%), Ara (15%), and GalA (38%) (Figure 4a). *luh* presented high Ara and low Rha and GalA levels (Figure 4a). *stk,* in the opposite manner, contained low Ara and higher WT Rha and GalA levels. The *luh stk* behaved in the same way as WT in Ara and GalA but had lower Rha levels (similar to *luh*) (Figure 4a). The levels of Xyl, Fuc, and GlcA were similar in all the genotypes, while Gal, Man, and Glu were higher in *luh stk* compared to WT and the single mutants (Figure 4a).

The GalA/Rha ratio represents the proportion between HG/RG-I. *luh* mutant seeds had a statistically different GalA/Rha ratio (an increase of approximately 50%) compared to WT seeds, while *stk* had a WT-like ratio (Figure 4b). The double mutant also possessed a statistically increased ratio with respect to the WT, which was similar to the single *luh* mutant, suggesting that the *LUH* mutation is epistatic over *STK* for the ratio of HG vs. RG-I and that *STK* disruption does not affect this ratio. Analysis of branching of RG-I levels based on Ara/Rha and Gal/Rha demonstrated opposite trends in ramification by *STK* and *LUH* mutations. *luh* pectins displayed a higher Ara/Rha ratio (280% increase), while *stk* seeds were characterized by a reduced Ara/Rha ratio (70% decrease) compared to WT (Figure 4b). We also found that the *luh stk* double mutant showed an increase, although minor, with respect to the single *luh* mutant in the Ara/Rha ratio (102%) compared to WT (Figure 4b). Calculations of Gal/Rha showed that *luh* CWs displayed a higher ratio, while *stk* presented similar values with respect to WT (Figure 4b). The *luh stk* double mutant had a slight increase in the Gal/Rha ratio with respect to the single *luh* mutant (Figure 4b). Finally, in order to analyze the possible impact of these mutants on the linearity of pectin, we calculated the ratio of the pectic backbone sugar GalA with respect to the neutral pectic sugars involved in side chains (GalA/FRAGX) [62]. We revealed an opposite trend among the two single mutants. *luh* pectins displayed a lower linearity ratio (13% decrease), while *stk* seeds were characterized by an increased ratio (13% increase) (Figure 4b) compared to WT. We found that the *luh stk* double mutant had a similar pectin linearity to WT (Figure 4b). 

These quantifications confirm that LUH and STK impact pectin components in different ways. LUH negatively controls RG-I branching and HG/RG-I ratios, while STK plays a minor role by positively controlling the RG-I ramification. 

Since LUH and STK act independently to induce size defects and to control pectin composition and pectin branching, we tested a possible interaction between these developmental regulators using a yeast two-hybrid assay (Y2H). Our analysis showed that LUH and STK were not able to interact (Figure 4c and Appendix A), supporting the idea that they act in different protein complexes to control CW homeostasis and pectin composition.

### 2.6. Structural Alterations of the Seed Coat Testa Cell Wall Influence Seed Germination

Multiple factors, such as molecular, hormonal, epigenetic, light quality, and abiotic stresses, regulate seed dormancy and germination [63,64,65]. In order to determine the effect of the CW-related genes on seed dormancy, germination of mutant seeds (freshly harvested and vernalized) was examined 24, 48, and 72 h after sowing under identical conditions (Table 1). 

As already described [28], *stk* mutants display early germination of the freshly harvested seed after 24 h when compared to WT, with a similar germination rate when observing vernalized seeds (Table 1). *luh* mutant seeds displayed faster germination than WT, showing 80% at 24 h for freshly harvested seeds and 60% for vernalized seed (Table 1). At 24 h, *luh stk* double mutant seeds had an accelerated germination rate of 84% for freshly harvested seeds, which is slightly faster than that of *luh* (80%). At 48 h, *luh stk* displayed a rate of 88% while *luh* reached 96% (Table 1), suggesting that absence of *LUH* and *STK* promotes rapid germination in freshly harvested seeds. In contrast, *GL2* mutation affects germination rate in an opposite way when compared to *LUH* and *STK*. In fact, freshly harvested seeds of *gl2.8* at 24 h showed only a 6% germination rate. Moreover, the *gl2.8* mutation also affected vernalized seed germination, showing 29% germination compared to 45% for WT at 24 h (Table 1). These data demonstrate that STK, LUH, and GL2 affect seed germination in opposite ways; while LUH and STK regulate negatively, GL2 displays a positive effect, regulating their downstream gene targets.

We also analyzed the effects of the mutation of cellulose related genes *FEI2.1* and *CESA5* [41,57]. Germination of *cesa5* seeds showed a slight delay in germination in both freshly harvested and vernalized seeds (8% and 31% before 24 h, respectively). On the other hand, the *fei2.1* mutant was earlier compared to WT (21% before 24 h in freshly harvested seeds compared to 12% WT). The germination rate of *fei2.1* vernalized seeds was 72% at 24 h, compared to 45% for WT (Table 1). Overall, these data show that disruption of CW structure in *cesa5 and fei2.1* influences seed germination processes, with CESA5 acting as positive regulator and FEI2 as negative regulator.

As for the CW pectin regulators, two mutant alleles of *FLY1* were analyzed. *fly1.1* freshly harvested seeds were able to germinate faster than *fly1.3* seeds (31% and 9% at 24 h, respectively), but no difference in vernalized seed germination was observed (Table 1). Compared to WT, *fly1.1* showed earlier germination in freshly harvested seeds at 24 h, while *fly1.3* displayed a similar germination rate to WT (Table 1). Vernalized seeds for both *fly1.1* and *fly1.3* displayed faster germination, reaching 90% and 92% at 24 h, respectively, while WT showed 45% (Table 1). We also used loss-of-function mutant lines of the *PMEI6* and *PER36* genes. Freshly harvested *pmei6* seeds were able to germinate at 48 h (75%) (Table 1). Mutations in *per36* severely affected seed germination in freshly harvested seeds, with no germination at 24 h. In contrast, vernalized *per36* seeds reached 68% before 24 h (exceeding WT, 45%) (Table 1). Therefore, it is likely that PER36 and PMEI6 affect seed coat structure in a way that impacts seed dormancy. Finally, to verify if hemicelluloses also impact seed germination, we tested the *bxl1* mutant. Its germination was delayed for both freshly harvested (0% at 24 h compared to 19% of the WS control) and vernalized seeds (0% at 24 h versus 39%), being able to germinate only at 48 h, which confirmed the importance of the CW structure in seed dormancy (Table 1).

In order to assess the importance of mucilage release, given the conflicting data regarding the positive or negative impact for the correct seed germination [66], we also analyzed the secretion of this pectic compound in mutants previously described. The formation of hydrophilic mucilage by the seed coat or pericarp, which is released upon seed hydration, is a commonly found adaptation in angiosperms, known as myxodiaspory. Upon the seed imbibition, mucilage begins to be extruded, creating a translucent halo around the seed which is composed by a non-adherent part and an adherent part. These are tightly attached to the seed itself, and are both characteristics of poorly branched RG-I pectin [23,67,68,69]. In this layer of mucilage, three major components are crucial for adherence: cellulose, hemicellulose, and pectins. The adherence of mucilage is controlled by complex interactions between the different CW polymers: cellulose, xylans, pectins, and glycoproteins [70]. Pectin staining with ruthenium red (RR) was used to visualize the extrusion of the different pectinaceous components on the mature seed epidermis (adherent and non-adherent mucilage).

Seeds from *cesa5* and *fei2.1* mutants released mucilage similarly to the WT, where the non-adherent and the adherent part of the mucilage consistently surrounded the seed (Appendix A). However, the cases of *cesa5* and *fei2.1* mucilage extrusion appeared to be “disorganized” compared to the WT, resembling the reduction in the adhesion of mucilage pectin as previously described [57] (Appendix A). In addition to cellulose, hemicelluloses are important for the mucilage structure. Mutation of *BXL1* leads to the production of seeds with a loss of mucilage extrusion, as already reported [40] (Appendix A). Mutation of *FLY1* showed slight differences in the phenotypes between the two alleles analyzed. *fly1.1* showed a reduction in mucilage extrusion with respect to WT (Appendix A). This reduction was more enhanced in *fly1.3,* where the RR coloration remained very close to the seed (Appendix A). Similarly to *fly1.1*, *per36* showed reduced mucilage extrusion compared to WT, while in the *pmei6* mutant, the mucilage secretion is completely abolished (Appendix A). These results are in line with what has already been observed in previous works [31,39,71,72].

Lastly, we analyzed mutants of transcriptional regulators known to be involved in CW remodeling. The three genotypes, *stk*, *luh,* and *gl2.8*, showed no mucilage release, as has already been reported in the literature [29,35,73]. The double mutant *luh stk* also failed to extrude mucilage (Appendix A).

### 2.7. Proanthocyanidins (PAs) Accumulation Is Not Affected in CW Related Mutants

The development of a proper seed coat is essential for seed embryos to cope with the surrounding environment [74]. In Arabidopsis seeds, PAs are synthesized and accumulated in the innermost layer of the seed coat, known as the endothelium [75]. Interestingly, different mutants defective in PAs synthesis showed smaller seeds with reduced length and width [75]. In order to understand whether PA accumulation is affected in the CW mutants, we performed a staining of this compound with vanillin. From our analysis, we found no significant differences between any of the different mutants compared to WT, except for *stk* and *luh stk* (Appendix A). The *STK* mutation leads to an ectopic accumulation of PAs in the seed coat, as previously reported and characterized [76,77]. We found that the *luh stk* double mutant seeds resembled the *stk* mutant phenotype presenting the “typical” ectopic layer of PAs, which appeared specifically in the third layer of the seed coat and, to a minor extent, in the second layer (Appendix A). These results suggest that, aside from *STK,* none of the other genes affecting CW structure in the seed coat are involved in PA accumulation. Furthermore, LUH does not interfere functionally on the action of STK in other cell layers of the seed coat.

## 3. Discussion

### 3.1. Cell Wall Processes Modulate Seed and Fruit Growth

The final seed size of angiosperms is a fine balance between endosperm expansion and seed coat extension [78]. Seed coat CW expansion coordinates plant organ shape. In this sense, the CW acts as an important physical factor, and its biophysical properties are a fundamental element of plant cell growth [79,80]. In order to understand how different polysaccharides affect seed shape, we measured the size of seeds belonging to CW-related loss-of-function mutants.

BXL1 is reported as a bifunctional β-d-xylopyranosidase/α-l-arabinofuranosidase [81]. *BXL1* is expressed in the vascular region of roots, leaves, flowers, and siliques [81], and it is positively regulated by LUH in the seed coat mucilage in seeds 7 days after pollination [35]. This expression pattern and the morphological defects observed in the mutants [81] implicate a possible role for BXL1 in seed development. The *bxl1* loss-of-function mutant showed smaller seeds compared to the corresponding WT (Ws) due to a dramatic reduction in width (Figure 1). This suggests that BXL1 promotes cell growth through its activity in hemicellulose loosening. In addition, hemicelluloses such as xyloglucan are able to form interconnections with cellulose microfibrils and reinforce the CW [82], which supports the *bxl1* seed phenotype. Further studies, which are out of the scope of this work, should be oriented towards determining whether the smaller seed size of *bxl1* is due to the seed coat phenotype, to embryonic/endosperm contribution, or to a combination of them.

PMEI, together with PMEs, play a role in CW reorganization [83]. The expression of *PMEI6* in epidermal cells of the seed coat inhibits PME activity on methylesterified HG in primary CW and mucilage [31]. Demethylesterification of pectin, caused by PME activity, can change the CW mechanical properties through the Ca^2+^ cross-links in pectin chains [84], and can promote the formation of ‘egg-box’ structures [85]. However, that process can cause different physiological effects that may increase or decrease CW stiffness [86]. Very little is known regarding the role of PMEs and the relative degree of methylesterification in the process of seed growth. It has been shown that modifications to the degree of pectin methylesterification through overexpression of a PMEI in Arabidopsis changed the mechanical properties of the micropylar endosperm and radicle cells [26]. In that work, it was reported that induced PME inhibition via *PMEI* overexpression produced the generation of bigger seeds. In agreement, we observed that *PMEI6* disruption reduces seed size, influencing both length and width (Figure 1). 

More recently, thanks to the functional characterization of the FLY1, it has been demonstrated that this ligase might use polyubiquitination to target PME proteins for degradation [39]. Here, we showed that the *FLY1* mutation, as well as *PER36*, negatively affect the seed size, demonstrating the importance of pectin maturation and mucilage extrusion factors in seed development (Figure 1).

Pectin can interact directly with cellulose microfibrils, suggesting a role for pectin in modulating the CW properties through interactions with cellulose [87,88]. When cellulose interacts with other CW polymers, it can be severed and formed in muro, limiting plant CW growth [82]. The amount of non-esterified HG increases to compensate for the lack of cellulose. This happens in response to a reduced cellulose synthase expression [89] or following treatment with cellulose synthesis inhibitors [18], suggesting that defective mutants in cellulose production may contain non-esterified HG. This could be in line with our observation of the *cesa5* phenotype, which, like *pmei6*, develops smaller seeds (Figure 1). In particular, *cesa5* seeds showed a reduction in length when compared to WT (Figure 1). This phenotype resembles, in part, what was previously seen for the *stk* mutant [28]. Interestingly, it was discovered that *CESA5* expression was downregulated in *stk* plants during seed development, with a consequent reduction in crystalline cellulose content [29]. STK likely influences cell length through an indirect or direct regulation of *CESA5* expression. Moreover, consistent with the *cesa5* “short seed size” phenotype shown here was the finding that CESA2, CESA5, and CESA9 are involved in radial CW reinforcement. A *cesa2 cesa5 cesa9* triple mutant was delayed in columella deposition, which may affect seed shape [90]. Genetic studies revealed that CESA5, CESA2, CESA6, and CESA9 are partially functionally redundant in primary CW biosynthesis [91,92]. In addition, *cesa9* also displayed smaller seeds [93], while a different allele of *cesa5* (*cesa5-1*) displayed seed size with no statistically differences compared to WT [94].

Both CESA5 and FEI2 (a receptor-like kinase) are involved in cellulose deposition in seed mucilage. CESA5 synthesizes cellulose to form primary CW, while FEI2, together with SALT OVERLY SENSITIVE 5 (SOS5), forms transverse rays in the inner layer of seed coat mucilage [41,57]. We demonstrated that *fei2* mutant plants show a reduction in seed and silique sizes (Figure 1), suggesting the positive role of FEI2 in both seed and silique growth, while CESA5 acts exclusively on seed growth.

### 3.2. SEEDSTICK and LEUNIG-HOMOLOG Contribute to Seed Development

We confirmed that *stk* develops smaller seeds; although smaller in length, they have a WT-like width (Figure 2). As shown recently, the development of smaller seeds upon *STK* mutation correlates with the direct transcriptional regulation of *XYL1* by this TF [28]. In addition, the STK-dependent control of seed size could be linked to a mechanistic effect of this TF. It has been demonstrated that the seed coat epidermal cells of the *stk* developing seeds are stiffer than those of the WT [29]. Moreover, STK is a positive regulator of *PMEI6* expression, inhibiting PME activity and controlling the seed coat strength. It has been well reported that PME negatively regulates CW growth [95]. Interestingly, STK and LUH negatively regulate one another in seed coat developmental pathways [29], but both mutants produce smaller seeds (Figure 2). LUH is also a positive regulator of *PMEI6* [31], and *luh* mutants contain high levels of methylesterified HG [10]. It is possible that LUH regulates other proteins involved in HG methylesterification, such as FLY1, which controls mucilage pectin demethylesterification [39], while it has previously been shown that *FLY1* expression is not affected in the *stk* mutant [29]. 

Intriguingly, we showed that the double mutant *luh stk* displayed a WT-like seed area, although this recovery was due to the increase in width compared to *stk* (Figure 2). This may indicate the existence of independent modes of action between these regulators of seed growth. A possible explanation would be that STK and LUH act in a different spatial way to positively control *PMEI6,* leading to specific and concrete growth defects. Supporting this dual role of STK and LUH over seed size, we have shown that the phenotypes of the single mutants are different, although they negatively impact seed size (thus, it seems that *STK* likely acts by controlling seed length, while *LUH* likely acts by seed width) (Figure 2). We also saw that the impact on total CW composition (Figure 3) and more specifically in isolated pectins (Figure 4) of both single mutants was different. The ramification sizes of RG-I, represented by the ratios of Ara and Gal (monosaccharides present in the side chains) relative to Rha residues (constituent of the RG-I backbone) were different between *stk* and *luh* (Figure 4). *luh* seeds are, apparently, more ramified in RG-I, while *stk* seems to have a very low RG-I ramification with respect to WT (Figure 4). Remarkably, these relative ratios were also reported to be low in the *pmei6-1* mutant whole CW extracts (10% decrease in both Ara/Rha and Gal/Rha, 4.02 and 2.87, respectively, vs. 4.46 and 3.16 in WT) [31]. The *pmei-6* mutant displayed smaller seeds due to a reduction in both seed length and width (Figure 1). Since STK is an activator of *PMEI6*, it is, therefore, reasonable to think that this genetic control imposes alterations to RG-I ramification. 

The values of HG/RG-I measured with the GalA/Rha ratio, for example, also showed that *luh* had pectins with proportionally more HG components relative to RG-I, with respect to WT (Figure 4). A high GalA/Rha rate (10% increase) was also reported in CW composition analysis performed in whole *fly1.1* seeds compared to WT (1.78 vs. 1.62, respectively) [39], which is interesting since we found that *fly1.1* seeds, as *luh*, had smaller seed areas due to a negative impact on seed width (Figure 1 and Figure 2). The *stk* seeds had WT-like GalA/Rha ratios (Figure 4); thus, it seems that the proportion of HG to RG-I is not affected upon STK disruption. In a similar way, intact GalA/Rha ratios were observed in *pmei6-1* mutant seeds with respect to WT seeds [31]. Since STK-PMEI6 controls PME activities and affects HG demethylesterification, these observations might suggest that PME activities affecting HG do not impact the distribution of major pectin groups (the ratio of HG/RG-I). 

Intriguingly, the double *luh stk* produced seeds with a “wrinkled” phenotype, and many seeds were characterized by epidermal cells with a depression in the center of the columella (Figure 2). This observation is similar to that of the strong allele already characterized (*luh*-*1*), but it was not observed for the other alleles already described in the literature (*luh-2* and *luh-4*) [10] and in this work (*luh-3*), which may suggest that *STK* disruption enhances the intensity of the *luh* phenotype. We hypothesize that this “wrinkled” phenotype of the double mutant derives from mutual and additive effects of STK and LUH on CW properties, endowed by each type of CW polymer, rigidity, flexibility, permeability, and resistance to desiccation. These developmental defects of seeds suggest that LUH and STK could act independently within the CW pathway, controlling seed development, as recently proposed [33]. In agreement, the Y2H assay performed here showed that LUH and STK proteins are not interacting (Figure 4 and Appendix A), suggesting that they act in different protein complexes controlling seed development.

Despite the possible role of LUH in seed growth, the silique of single mutant *luh*, which had the same length with respect to WT, suggests that this transcriptional co-regulator does not have any role in fruit growth.

Neither *luh* nor *stk* single mutants produce mucilage after imbibition [10,29,35] (Appendix A). Changes in mucilage extrusion have been linked to methylesterification of HGs [38]. Higher levels of methylesterified HG were identified in *luh* CWs [10,73], while highly methylesterified HGs were absent in *stk* mutants [29]. The difference in methylesterified HGs may be the cause of the germination effects observed here. *luh* was able to germinate earlier in both freshly harvested and vernalized seeds compared to *stk,* which was able to germinate faster, when we considered freshly harvested seeds (Table 1) as already described in the literature [28]. *luh stk* seeds also displayed the same pattern observed in *luh*. However, the difference in HGs needs to be clarified since LUH and STK both positively regulate *PMEI6* [29,31]. 

*GL2* encodes a TF that is expressed in the outer and inner integument, which is required for seed coat mucilage biosynthesis, at least in part by the positive control of the *MUM4*/*RHM2* gene [96,97,98,99]. The activity of GL2 in early stages of seed development, specifically in the outermost cell layer of the outer integument, as well as the reduction in seed size due to the reduced length and width of *gl2.8* seeds (Figure 1), supports the role of GL2 in seed development. It has been observed that GL2 affects neither the expression of *LUH* [35] nor *STK* [29], but, in an opposite manner, positively controls the expression of other seed coat CW regulators such as *PMEI6*, *MUM2*, *BXL*, and *SBT1.7*. LUH positively controls *PMEI6* and negatively controls *STK*, while STK positively controls *PMEI6* but negatively controls *LUH* [29,31]. Therefore, the results obtained here would indicate that GL2’s control over seed size is due to effectors other than STK and LUH.

### 3.3. Cell Wall Processes Modulate Seed Germination

Plants from many species produce mucilage polysaccharides, which may facilitate seed germination, and confer the major adaptive advantage commonly found in angiosperms [66,100]. It has been suggested that the seed coat confers a key structure required for germination. In agreement with this, we previously found that changes in germination of *stk* are related to the seed coat structure, which involve water absorption, alterations in flavonoid pigmentation and PAs contents, free xyloglucan oligosaccharides, and accessible polymeric xyloglucan, as well as cell wall structural and mechanical defects [28,29]. In addition, the other two transcriptional regulators used in this study have already been proposed as important to the germination rate. Indeed, the *gl2*.*8* mutant has a clear delay in germination considering both tests (freshly harvested and vernalized seed, Table 1) at the three time points analyzed, as already described [75]. Concerning the impact of the *LUH* mutation on germination rate, it was previously reported that the *luh-1* allele, after sowing on MS-plate, displayed slowed germination [101]. In this work, we have shown that the *luh-3* allele displayed an increase in germination rate, particularly after 24 h and 48 h, with respect to WT considering the freshly harvested seed, while vernalized seeds had a rapid germination rate at 24 h from sowing (Table 1). As to these contrasting results, we could not exclude that the different conditions of the experiments, or the different position of the mutations for the two alleles, could explain the dissimilar results obtained in this work. In agreement with this, as previously mentioned, the columella defects of the strong *luh-1* allele and the ones described in this work (*luh-3*, Figure 2 and Appendix A) appear different. Nevertheless, LUH acts as an important co-regulator of germination by controlling Phytochrome-interacting factors 1 (PIF1) [102]; therefore, we cannot exclude differential impact of this important phytohormone regulator in the different alleles leading to different germination dynamics.

All the single mutants for the three transcriptional regulators analyzed in this work showed a loss of mucilage extrusion (Appendix A), which is not correlated with a similar delay or faster germination with respect to WT (Table 1). These observations corroborated what has been previously hypothesized regarding the unclear correlation between mucilage extrusion and germination [66]. 

FLY1 regulates pectin methylesterification by recycling PMEs. In fact, high levels of demethylesterified pectin in *fly1* mucilage were reported [39]. That finding could explain the rapid *fly1.1* seed germination in our tests, which reached more than 90% before 48 h for freshly harvested seed, and earlier at 24 h for vernalized seed of both *fly1.1* and *fly1.3* alleles (Table 1). This is likely due to the pectin-altered composition that made it easier to break the seed testa. Interestingly, at 24 h after sowing, the freshly harvested seeds, the two *fly1* alleles demonstrate opposite behaviour. *fly1.1’s* germination at 24 h is faster than that of WT, while the *fly1.3* is slower (Table 1). This slight difference is also reflected in mucilage extrusion, in which *fly1.3* showed a reduced extrusion compared to *fly1.1* (Appendix A).

Evidence that pectin biosynthesis and maturation are important for seed germination was also reported in other species. Tissue-specific pectin methylesterification and PME activities play a major role in lettuce seed germination [103]. In Arabidopsis, it was reported that induced *PMEI* overexpression led to faster rates of seed germination [26], while we have revealed in our work that disruption of *PMEI6* delays germination (Table 1) with a loss of mucilage release (Appendix A) [31]. Since *pmei6* mutants presented a reduced germination rate compared to *luh*, *stk*, (as well as to WT, Table 1), it would seem reasonable to believe that HG methylation does not play a major role (if any) in the germination defects (anticipated) observed in *luh* and *stk*.

We found that *per36* exhibited a delay in germination, reaching 57% at 72 h after sowing, making it one of the most delayed germinators under our experimental conditions (considering the freshly harvested seeds) (Table 1). The mucilage extrusion of the *per36* mutant was characterized by a partial release, as has already been reported and confirmed in our work (Appendix A) [71]. Finally, the mutation of the hemicellulose-related gene *BXL1* displayed delayed germination, particularly for the freshly harvested seeds (Table 1), as already described [40]. Moreover, *bxl1* mutant seeds do not extrude mucilage (Appendix A). 

It seems that these changes in mucilage extrusion observed in mutants affecting CW components, such as *pmei6*, *per36*, *fly1.3,* and *bxl1,* seeds are determinant in controlling seed germination, specifically in freshly harvested seeds, which more closely resemble natural environmental conditions. Interestingly, these differences were not observed for seeds in vernalized conditions. Our observations corroborate the proposed role of mucilage, which possesses hydrogel properties, in enhancing seed water uptake during the imbibition and regulation of seed germination [104]. 

## 4. Materials and Methods

### 4.1. Plant Material

Seeds were sterilized with 70% ethanol for 2 minutes (m) and 1% bleach for 5 m, then were washed three times with sterile water. Seeds were then sown in MS medium [105] and germinated in petri dishes in a growth chamber (25 °C and 16 h of light). After 4 days, the seedlings were moved to soil and grown in a greenhouse (25 °C and 16 h of light). T-DNA insertional loss-of-function mutants were screened using PCR, with specific oligonucleotides following indications on previous references, and were consequently phenotyped (details of the seed stocks’ origins are described in Appendix A). All the mutants used in this work were in Columbia (Col-0) background, except for *bxl1,* which was obtained in Wassilewskija (Ws).

### 4.2. Seed and Silique Morphological Characterization 

For seed measurements, images were taken using a Leica MZ6 stereomicroscope with 5 biological replicates and 50 seeds each. The seeds were collected from a pool of 5 independent plants. The images were analyzed using the SmartGrain software [106], given seed size (area size-AS) (refers to the total contents within the border), length (L) (longitudinal length), and width (W) (transverse to longitudinal length). For silique length, photos were taken using a stereomicroscope with 3 biological replicates of 10 siliques each. The siliques were collected from a pool of 5 plants. The photos were analyzed with ImageJ software [107]. Statistical analyses were performed using ANOVA, followed by Tukey’s honestly significant difference (HSD) test (*p* < 0.01).

### 4.3. Germination Test

Seeds from WT and mutant plants were grown side-by-side under identical environmental conditions. In order to verify the effects of dormancy, freshly harvested seeds (seeds harvested from the plants and immediately used for the experiment) and vernalized seeds (seeds harvested and stored for 6 weeks) were used. In order to overcome dormancy, seeds were kept at 4 °C for 7 days [108]. Seeds were surface sterilized and placed in MS medium [105] without sucrose in a growth chamber (25 °C and 16 h of light). Seeds were observed 24, 48, and 72 h after sowing, and the experiment was performed with 50 seeds for each mutant and corresponding WT (similar results were obtained with seed stocks from a second set of plants growing independently). Seeds were scored as germinated when testa rupture preceding radicle protrusion was visible. Statistical analysis and analysis of variance (ANOVA) were performed using SAS 9.3^®^ software (https://www.sas.com, accessed on 13 October 2021). To verify whether the germination of CW mutants was significantly different from WT, means were compared by the Dunnett’s test with *p* ≤ 0.05.

### 4.4. Yeast Two-Hybrid

Yeast two-hybrid (Y2H) assays were performed using the GAL4 system, as previously described [109,110]. pDEST22 allowed for GAL4 AD fusion with the N-terminus region of the LUH and STK proteins, while pDEST32 allowed for GAL4 BD fusion with the N-terminus region of SEP3. For the Y2H assay, BD-STK was first tested for autoactivation, followed by a Y2H test with AD-SEP3 [109] and AD-LUH clone from the EU-REGIA project [111]. The final scoring was performed seven days after incubation at 22 °C.

### 4.5. Cell Wall Extraction and Pectin Enrichment Analyses from Mature Arabidopsis Seeds

Seeds of Arabidopsis WT, *luh*, *stk,* and *luh stk* mutants were suspended in 70% ethanol at 70 °C, ground to a fine powder in a mortar for 3 m and incubated for 15 m under vigorous agitation in a 70 °C water bath. Insoluble residues were collected after centrifugation (10 m, 4000× *g*). Ethanol extraction was repeated twice. A series of extractions was then performed to remove lipids, polyphenols, and low molecular mass metabolites from the cell wall residues. Briefly, the residues were extracted with methanol–chloroform (1:1; *v/v*), then with methanol–acetone (1:1; *v/v*), and, finally, with acetone for 2 h each at room temperature under agitation. Insoluble residues were then air flush dried. This total CW fraction was called alcohol insoluble residue (AIR). The experiment was performed with 4 biological replicates of seed pools obtained from 5 plants of each mutant and WT, grown separately. For pectin enrichment analysis, CW material (10 mg AIR) was treated with 4 mL of 0.1 M boiling ammonium oxalate for 1 h under vigorous shaking. After a 4000× *g* centrifugal separation, solubilized material was collected from supernatant. The pectin extraction was repeated once. The pectin-enriched fractions were dialyzed (3.5 kDa exclusion size) against deionized water for two days using Spectra/Por2 Dialysis tubing membranes (https://www.fishersci.com, accessed on 19 April 2022). Pectin material was lyophilized.

### 4.6. Monosaccharide Composition Analysis

Monosaccharide composition of each AIR was analyzed by gas chromatography coupled to a Flame Ionization detector (GC-FID) spiking inositol as an internal standard. Each fraction (1 mg) was hydrolyzed with 2 M trifluoroacetic acid (TFA) for 2 h at 110 °C. TFA was rinsed twice with a 50% isopropanol:water washing solution. The released monosaccharides were converted to their O-methyl glycosides by incubation in 1 M methanolic HCl at 80 °C overnight [112]. After evaporation of methanol, the methyl glycosides were then converted into their trimethylsilyl derivatives by heating the samples for 20 m at 110 °C in hexamethyldisilane-trimethylchlorosilane-pyridine (3/1/9). After evaporation of the reagent, the samples were suspended in cyclohexane before being injected on a CP-Sil 5 CB Agilent Technologies. Chromatographic data were integrated with OpenLab software (Agilent, Santa Clara, CA, USA, https://www.agilent.com, accessed on 22 March 2022). A temperature program (3 m at 40 °C; up to 160 °C at 15 °C m^−1^; up to 220 °C at 1.5 °C m^−1^; up to 280 °C at 20 °C m^−1^; 3 m at 280 °C) optimized for the separation of the most common CW monosaccharides chemically derivatized, such as arabinose (Ara), fucose (Fuc), galactose (Gal), galacturonic acid (GalA), glucose (Glc), glucuronic acid (GlcA), mannose (Man), rhamnose (Rha), and xylose (Xyl). Ara/Rha and Gal/Rha ratios can reflect the relative importance of Ara or Gal in side-chains of RG-I, while the GalA/Rha ratio reflects the proportion between RG-I and HG, as these two monosaccharides are unique monomers of these respective pectin backbones [113,114,115]. The last sugar ratio reported here described the pectic backbone sugar GalA with respect to the neutral pectic sugars involved in side chains (GalA/FRAGX), and this inferred a measure for the linearity of pectin [62]. These ratios were determined for every fraction and biological sample, and average values were presented. 

### 4.7. Proanthocyanidin and Mucilage Extrusion Analyses 

In order to check the PA accumulation, fresh seeds from silique at 3 days after pollination have been collected. Slides were mounted with a solution made by 1% (*w/v*) vanillin (Sigma-Aldrich, St. Louis, MI, USA, https://www.sigmaaldrich.com, accessed on 18 May 2021) and 5 M HCl and incubated at room temperature for 5 m. Samples were observed using a Zeiss Axiophot D1 microscope.

To check the mucilage extrusion, freshly harvested seeds were incubated for 15 m with a solution of 0.01% (*w/v*) ruthenium red (Sigma-Aldrich, https://www.sigmaaldrich.com, accessed on 18 May 2021). Samples were then analyzed using a Zeiss Axiophot D1 microscope.

## 5. Conclusions

The influence of the seed coat on seed and fruit development has been well demonstrated, particularly the molecular control of the composition of the epidermal CW [26,28,29,30,116]. We analyzed the morphological and physiological effects of loss of function mutations in genes involved in cellulose, hemicellulose, and pectin. We proposed a model wherein many of the seed coat developmental players control development of seeds and germination (Figure 5). Germination is a critical stage in the life of a plant and constitutes a remarkable checkpoint for plant survival. Understanding germination can help to improve seed priming (regulated germination) and the possible coexistence with other crops [117]. Our results suggest that alterations in pectin components such as RG-I branching may impact seed growth and germination. We believe that the Ara or Gal side chains of RG-I play a critical role in the ability of CWs to remain flexible during plant growth and may have important functions in relation to the absorbance of water content upon seed imbibition. The importance of these ramifications has recently been linked to salt tolerance via hydration of the seed endosperm during germination [118], and to the induction of tolerance in seeds to dehydration [119]. Further studies on the relationship between RG-I side chains and CW flexibility may provide novel perspectives to understand the role of these polysaccharides in the life of the plant.

## Figures and Tables

**Figure 1 plants-11-03146-f001:**
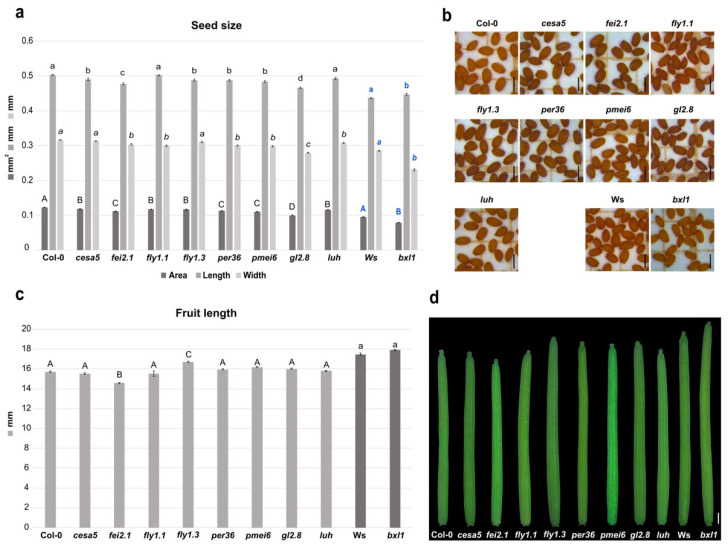
Effects of the cell wall mutations in seed and silique development. (**a**) Histogram representing seed size of the cell wall mutants, with mean of the seed area (mm^2^), length (mm), and width (mm); Error bars indicate the standard error (SE) of one representative biological replicate of *n* = 50 seeds. Statistical analyses were performed using ANOVA, followed by Tukey’s HSD test. Different letters indicate statistically significant differences (*p* < 0.01) (WT versus the other genotypes). (**b**) Stereomicroscope images of seeds from the cell wall mutants. Scale bar = 0.5 mm. (**c**) Histogram representing the mean of the silique length of the cell wall mutants; error bars indicate the SE of one representative biological replicate of *n* = 10 siliques. Statistical analyses were performed using ANOVA followed by Tukey’s HSD test; different letters indicate statistically significant differences (*p* < 0.01) (WT versus the other genotypes). (**d**) Stereomicroscope images of the siliques from cell wall mutants. Scale bar = 1 mm.

**Figure 2 plants-11-03146-f002:**
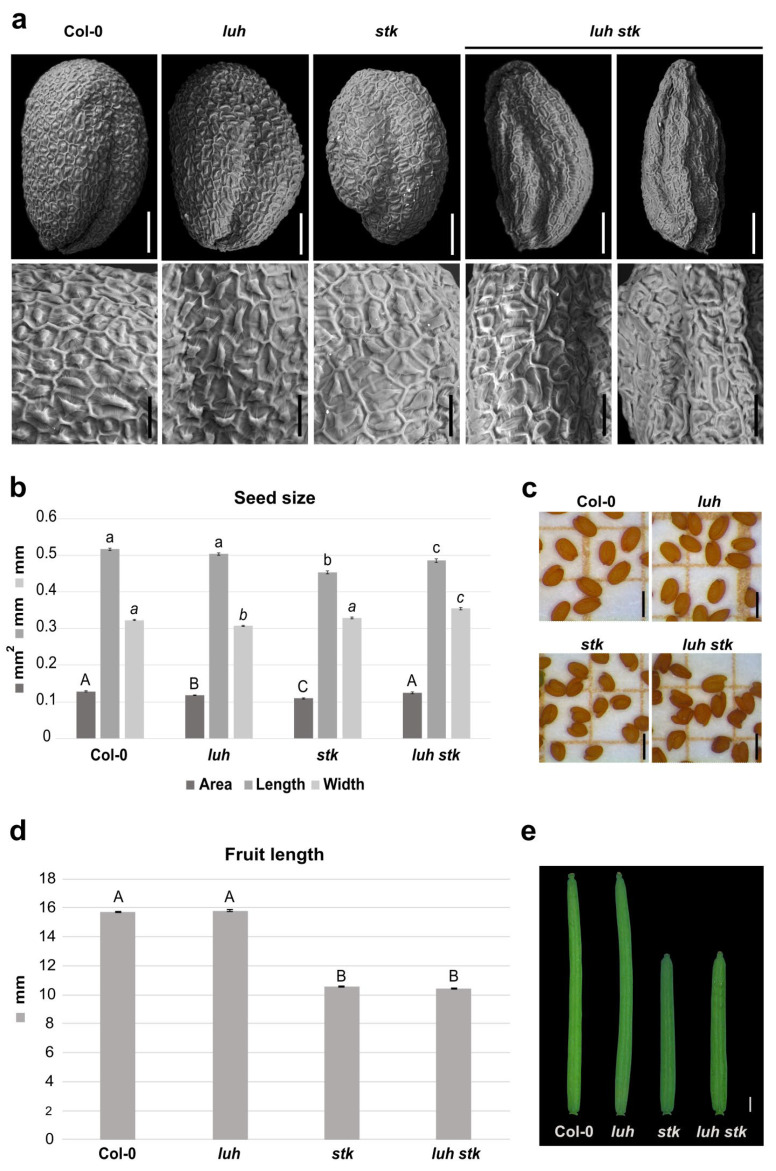
Morphological effects of *luh stk* double mutant on seed and silique development. (**a**) Scanning Electron Microscopy (SEM) pictures of seeds from Col-0 (WT), *luh*, and *stk* single mutants as well as *luh stk* double mutant. In the upper part, the whole seeds are shown, scale bar = 100 µm; in the bottom part, pictures with a zoom of the seed surface are shown, scale bar = 20 µm. (**b**) Histogram representing the seed size of WT, *luh*, *stk,* and *luh stk,* with mean of the seed area (mm^2^), length (mm), and width (mm). Error bars represent the SE of one representative biological replicate of *n* = 50 seeds. Statistical analyses were performed using ANOVA followed by Tukey’s HSD test. Different letters indicate statistically significant differences (*p* < 0.01) between WT and the other genotypes, and between single mutants and the double mutant. (**c**) Stereomicroscope images of seeds from WT, *luh*, *stk,* and *luh stk*. Scale bar = 0.5 mm. (**d**) Histogram representing the silique length of WT, *luh*, *stk,* and *luh stk*. Error bars represent the SE mean of one representative biological replicate of *n* = 10 siliques for each genotype. Statistical analyses were performed using ANOVA followed by Tukey’s HSD test. Different letters indicate statistically significant differences (*p* < 0.01) between WT and the other genotypes and between single mutants and the double mutant. (**e**) Stereomicroscope images of silique from WT, *luh*, *stk,* and *luh stk*. Scale bar = 1 mm.

**Figure 3 plants-11-03146-f003:**
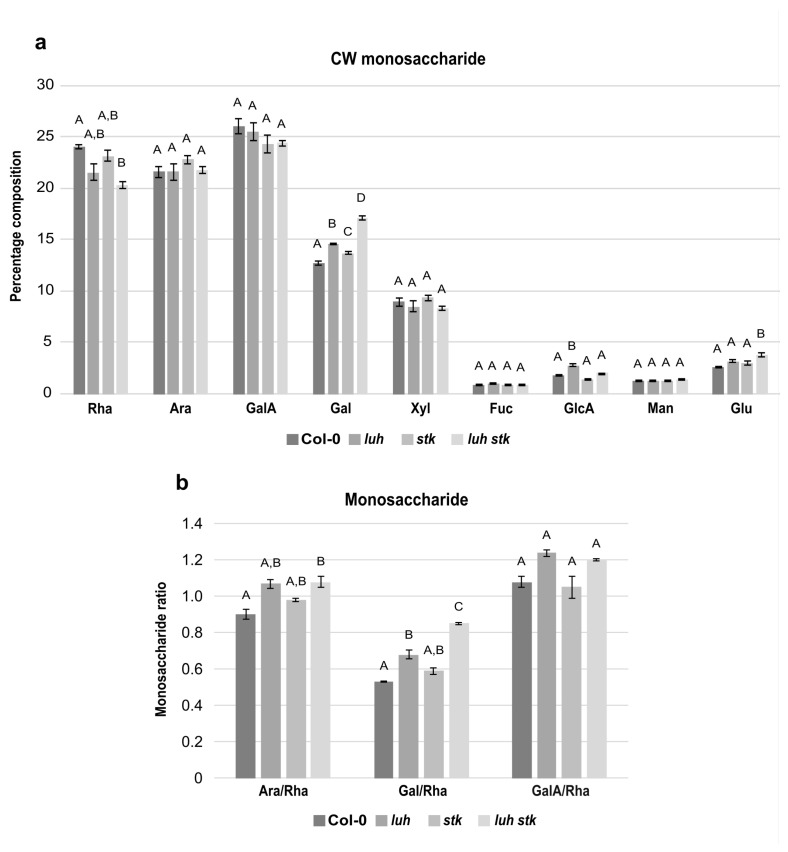
Total cell wall monosaccharide composition in seeds from Col-0 (WT), *luh* and *stk* single mutants, as well as the *luh stk* double mutant. (**a**) Monosaccharide composition. Rha, rhamnose; Ara, arabinose; GalA, galacturonic acid; Gal, galactose; Xyl, xylose; Fuc, fucose; GlcA, glucuronic acid; Man, mannose; Glc, glucose. Statistical analyses were performed using ANOVA followed by Tukey’s HSD test; different letters indicate statistically significant differences (*p* < 0.01) between WT and the other genotypes, as well as between single mutants and the double mutant. (**b**) Pectin distribution and ramification. Ara/Rha ratio, Gal/Rha ratio, GalA/Rha ratio. Error bars represent the SE of 4 biological replicates. Statistical analyses were performed using ANOVA followed by Tukey’s HSD test; different letters indicate statistically significant differences (*p* < 0.01) between WT and the other genotypes, as well as between single mutants and the double mutant.

**Figure 4 plants-11-03146-f004:**
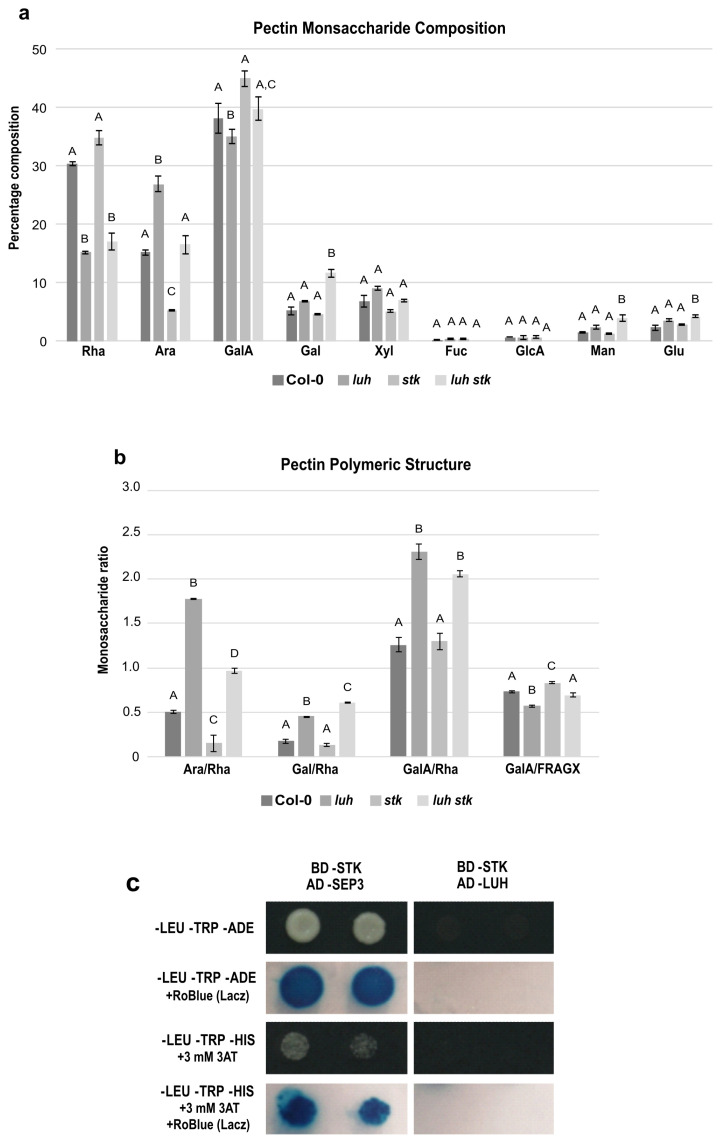
Pectin-enriched monosaccharide composition analysis of seeds from Col-0 (WT), *luh* and *stk* single mutants and *luh stk* double mutant. (**a**) Monosaccharide composition. Rha, rhamnose; Ara, arabinose; GalA, galacturonic acid; Gal, galactose; Xyl, xylose; Fuc, fucose; GlcA, glucuronic acid; Man, mannose; Glc, glucose. Statistical analyses were performed using ANOVA followed by Tukey’s HSD test. Different letters indicate statistically significant differences (*p* < 0.01). (**b**) Pectin distribution and ramification. Ara/Rha ratio, Gal/Rha ratio, GalA/Rha ratio, and GalA/FRAGX (fucose, rhamnose, arabinose, galactose, xylose). Statistical analyses were performed using ANOVA followed by Tukey’s HSD test. Error bars represent the SE of 4 biological replicates. Different letters indicate statistically significant differences (*p* < 0.01). (**c**) Yeast two-hybrid assay to test the LUH–STK protein interaction.

**Figure 5 plants-11-03146-f005:**
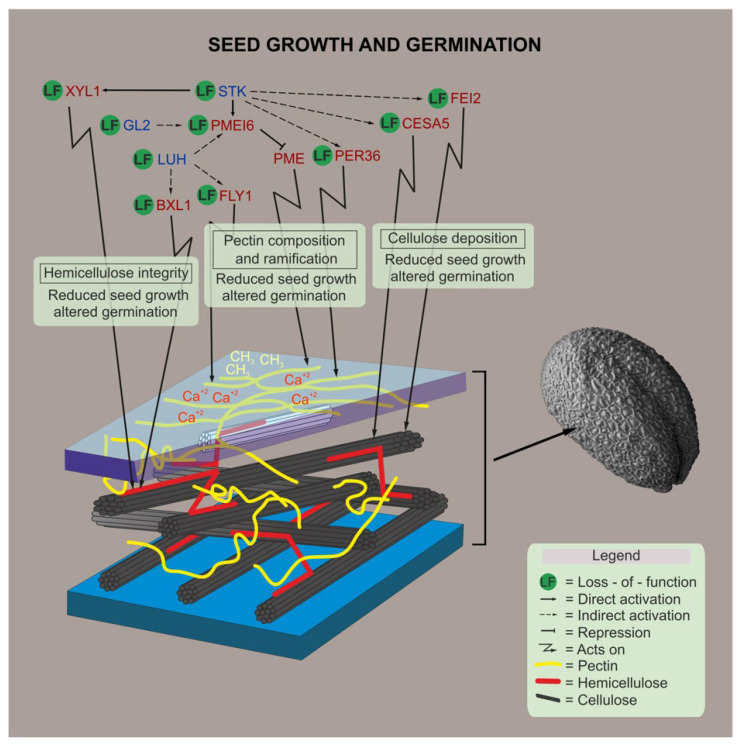
Proposed model. A schematic representation of the genes involved in CW polysaccharide remodeling, and how their mutations affect seed development and germination, are shown based on results presented in this work, as well as on genetic and biochemical evidences shown previously in the literature [28,29,31,35,57]. Blue letters indicate the transcriptional regulators, red letters indicate proteins with enzymatic activity, and dark blue indicates a protein without enzymatic activity.

**Table 1 plants-11-03146-t001:** Effects of the cell wall mutations in seed germination tested in freshly harvested and vernalized seeds over 24, 48, and 72 hours (h). Col-0 is the correspondent WT for all mutants and Wassilewskija (Ws) as the correspondent WT of *bxl1*. Standard Deviations (SD) are representative of 3 biological replicates.

Germination Test													
Type	Genotype	Freshly Harvested Seeds						Vernalized Seeds					
		24 h		48 h		72 h		24 h		48 h		72 h	
		%	SD	%	SD	%	SD	%	SD	%	SD	%	SD
Wild-type	Col-0	12.7	2.5	75.3	2.5	99.3	1.1	45.0	1.7	99.0	1.7	100	0
	Ws	19.3	3.2	50.3	2.5	70.7	1.5	39.7	1.5	59.7	4.2	88.7	2.5
Transcriptional regulators	*stk*	23.7 *^,+^	0.6	54.7 *^,+^	1.5	90.7 *	2.1	41.0 ^+^	3.0	76.7 *^,+^	1.5	100	0
	*luh*	80.0 *	2.0	96.3 *^,+^	3.8	100 ^+^	0	60.0 *^,+^	1.7	98.0 ^+^	1.0	100	0
	*luh stk*	84.0 *	5.3	88.3 *	0.6	92.0	4.6	83.7 *	5.8	99.3	1.1	99.7	0.6
	*gl2.8*	6.0 *	2.0	52.0 *	1.0	92.0	2.0	29.7 *	1.5	53.0 *	2.0	75.7 *	1.5
Cellulose biosynthesis	*cesa5*	8.7	1.1	73.3	3.5	100	0	31.3 *	1.1	89.3 *	1.1	97.7	2.5
	*fei2*	21.7 *	2.3	95.0 *	1.0	98.0	2.0	72.0 *	5.0	99.3	0.6	99.7	0.6
Pectin biosynthesis and maturation	*pmei6*	0 *	0	75.3	1.1	81.3 *	6.1	56.0 *	1.0	90.0 *	2.0	97.3	3.0
	*fly1.1*	31.7 *	1.5	91.7 *	1.5	99.0	1.7	90.3 *	1.5	99.3	1.1	100	0
	*fly1.3*	9.3	4.2	72.3	6.5	76.0 *	7.5	92.0 *	2.0	98.0	2.0	100	0
Mucilage extrusion	*per36*	0 *	0	18.3 *	2.1	57.0 *	3.0	68.0 *	1.7	86.7 *	6.1	93.3 *	2.1
Hemicellulose	*bxl1*	0 *	0	16.7 *	2.1	28.0 *	2.0	0 *	0	100 *	0	100*	0

* Different from its corresponding WT by Dunnett’s test (*p* ≤ 0.05). ^+^ Different from the *luh stk* double mutant by Dunnett’s test (*p* ≤ 0.05).

## Data Availability

The data presented in this study are available on request from the corresponding author.

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
