# Peer review of "The Genetic Control of SEEDSTICK and LEUNIG-HOMOLOG in Seed and Fruit Development: New Insights into Cell Wall Control"

_plants, 2022, doi:10.3390/plants11223146_

Round 1
Reviewer 1 Report
This manuscript reports a potpourri of seed coat cell wall defects from multiple relevant mutants in Arabidopsis. The research is carefully performed and the story is well written. I only have a few concerns that need to be addressed.
1. It’s disappointing that STK and LUH are not interacting with each other. Is Y2H the only method that has been tried?
2. It appears STK and LUH act independently, the authors should add some context when they say one’s mutation is dominant over the other one, such as line 277 and line 328.
3. In lines 87-88, LUH is described as a transcriptional repressor, while in lines 172 LUH is an activator. The authors should distinguish between molecular function and developmental role of LUH.
4. In lines 209-210, I’m not very persuaded that luh mutant has ‘regular’ cell shape, since it shows some similarity with the left photo of luh stk double mutant in Fig S1.
5. In Fig 2a, the double mutant seed appears a bit narrower than WT and single mutants, opposite to Fig 2b.
6. Line 489, I’m not sure if ‘antisense plant’ is a good expression here. The authors should add some explanation.
7. Generally, the discussion section appears too wordy. A compact and cohesive discussion is expected. In addition, I’m assuming that seed size and germination rate are related to cell wall stiffness. Do the authors have any clue that the mutants have altered cell wall rigidity? This point should be added to the discussion.
Author Response
We want to kindly thank the reviewer 1 for the suggestions and constructive comments for our manuscript.
This manuscript reports a potpourri of seed coat cell wall defects from multiple relevant mutants in Arabidopsis. The research is carefully performed, and the story is well written. I only have a few concerns that need to be addressed.
- It’s disappointing that STK and LUH are not interacting with each other. Is Y2H the only method that has been tried?
- We made this interaction only with the Y2H assay, and considering the results obtained in this work, for example what is described in the section 2.5, LUH and STK act independently to induce size defect. it is reasonable that LUH and STK do not interact.
- It appears STK and LUH act independently, the authors should add some context when they say one’s mutation is dominant over the other one, such as line 277 and line 328.
- We agree with the reviewer comment. We changed “dominant” to “epistatic” in order to make more clear the relationship between the two mutations. (Please see lines: 308,309).
- In lines 87-88, LUH is described as a transcriptional repressor, while in lines 172 LUH is an activator. The authors should distinguish between molecular function and developmental role of LUH.
- We have added in the line 175 the term “developmental” to distinguish the developmental role of LUH from the transcriptional function described in the introduction section.
- In lines 209-210, I’m not very persuaded that luh mutant has ‘regular’ cell shape, since it shows some similarity with the left photo of luh stk double mutant in Fig S1.
- To clarify this aspect, we have added in the Figure 2a also the SEM detailed pictures of the epidermal cell surface which were previously in the Figure S1. The luh single mutant presented here does not show an irregular epidermal surface, which appears like the WT. The double mutant has a clear depression at the center of columella which is not observed in both single mutants.
- In Fig 2a, the double mutant seed appears a bit narrower than WT and single mutants, opposite to Fig 2b.
- We agree with this comment, we have included a new Supplemental Figure (Figure S1), with SEM pictures in which you can appreciate that not all the seeds of the double mutant are characterized by a wrinkled phenotype. We also added in the new version, a clarification, in lines: 208,209 of the % of “wrinkled” and “non-wrinkled” phenotypes.
- Line 489, I’m not sure if ‘antisense plant’ is a good expression here. The authors should add some explanation.
- We agree with this comment, and we decided to remove this sentence.
- Generally, the discussion section appears too wordy. A compact and cohesive discussion is expected. In addition, I’m assuming that seed size and germination rate are related to cell wall stiffness. Do the authors have any clue that the mutants have altered cell wall rigidity? This point should be added to the discussion.
- We reduced the information described in the discussion section as the reviewer suggests. Moreover, the last paragraph 3.4. “Future perspectives in seed development” is reduced in the new version and it becomes the “Conclusion” section of the manuscript as the guidelines of the journal suggest.Moreover, we have added the information regarding the cell stiffness of stk seeds in the Discussion lines: 513, 514. There are not data regarding cell wall rigidity in seeds of luh mutants in literature.
Reviewer 2 Report
This manuscript attempts to provide new insights into the regulation of cell walls during seed development. Although the topic is of high interest, the article has several serious flaws and the results do not support the conclusions:
1. Most of the data presented in this study relate to seed dimensions, silique length, and whole seed monosaccharide composition. I identified numerous problems with presentation and interpretation of the phenotypic results. Many of the described changes (e.g. smaller seeds for cesa5 mutant) are inconsistent with other publications on the same mutants that are not cited. The magnitude of the described changes is not mentioned in the text and appear to be minor in most cases (e.g. <5% difference from wild-type for cesa5 in Figure 1). Some examples of references omitted:
Griffiths et al 2016. Dissecting Seed Mucilage Adherence Mediated by FEI2 and SOS5. Frontiers in Plant Science 7: 1–13.
Griffiths et al 2014. SALT-OVERLY SENSITIVE5 mediates arabidopsis seed coat mucilage adherence and organization through pectins. Plant Physiology 165: 991–1004.
Voiniciuc et al 2015. Highly Branched Xylan Made by IRREGULAR XYLEM14 and MUCILAGE-RELATED21 Links Mucilage to Arabidopsis Seeds. Plant physiology 169: 2481–95.
Yang et al 2020. Seed hemicelluloses tailor mucilage properties and salt tolerance. New Phytologist 229: 1946–1954.
Insufficient evidence is provided to conclude that the described genes (e.g. FLY1, CESA5, GL2) act as positive or negative regulators of seed development beyond what was previously published. Some inaccurate remarks are also present. Example: “mucilage is composed of a polar glycoprotein”. Most of Arabidopsis mucilage is pectin (specifically RG-I). No conclusive evidence is provided to support changes in RG-I branching (the sugar ratios are not enough).
2. The article is unusually long (18 pages before the methods) for the limited content of new results and the results/discussion text is too descriptive. For example, two pages of text describe the results of a single table (Table 1, section 2.7). The manuscript presents an unusually large amount of information that was previously known (example: Figure S3 and lines 421 to 455). In addition, most of the discussion reads like a broad literature review (e.g. section 3.1), rendering the overall scope of the manuscript ambiguous.
It may be necessary to split the manuscript in two: One as a broader scope literature review, and another focused on gaining new insights into cell wall regulation during seed development (incorporating additional experiments and a more balanced interpretation).
3. The data presentation must be improved. Fig. 1 and Fig. 2b graphs need separate Y-axes with their own values/units for the length/wide and area measurements. None of the figures in the supplementary information have legends. Table S1 has multiple references misspelled
For Table 1, information about the replicates for the SD calculation is needed; “freshly harvested” should be defined (in terms of days or weeks post-harvest). The text in this figure is also too small.
Minor points:
Line 243 – the three mentioned sugars would add up to more than 20-25% of total monosaccharides
“Pectinic” is mentioned several times starting line 316 and should be changed to pectic
Line 328 – “confirms” is too strong; “suggests”
Author Response
We want to kindly thank the reviewer 2 for the suggestions and comments.
This manuscript attempts to provide new insights into the regulation of cell walls during seed development. Although the topic is of high interest, the article has several serious flaws and the results do not support the conclusions:
- Most of the data presented in this study relate to seed dimensions, silique length, and whole seed monosaccharide composition. I identified numerous problems with presentation and interpretation of the phenotypic results. Many of the described changes (e.g. smaller seeds for cesa5 mutant) are inconsistent with other publications on the same mutants that are not cited. The magnitude of the described changes is not mentioned in the text and appear to be minor in most cases (e.g. <5% difference from wild-type for cesa5 in Figure 1). Some examples of references omitted:
Griffiths et al 2016. Dissecting Seed Mucilage Adherence Mediated by FEI2 and SOS5. Frontiers in Plant Science 7: 1–13.
Griffiths et al 2014. SALT-OVERLY SENSITIVE5 mediates arabidopsis seed coat mucilage adherence and organization through pectins. Plant Physiology 165: 991–1004.
Voiniciuc et al 2015. Highly Branched Xylan Made by IRREGULAR XYLEM14 and MUCILAGE-RELATED21 Links Mucilage to Arabidopsis Seeds. Plant physiology 169: 2481–95.
Yang et al 2020. Seed hemicelluloses tailor mucilage properties and salt tolerance. New Phytologist 229: 1946–1954.
Thanks for the comments. We have performed another time the statistical analyses to make sure that also the small differences in size were statistically significant. The new statistical analyses confirm that there were differences in Area size and length for example for cesa5 mutant seeds with respect to WT. The statically tests were described in the legend and in the materials and methods section. Moreover, we considered these interesting references, and we have added it in the new version of the manuscript:
- (Griffiths et al2014 and 2016) We added these references in the introduction section; (Please see lines: 101);
- (Voiniciuc et al2015) We inserted this reference in the new version, lines: 406, 407;
- (Yang et al 2021 lines 500, 501); In this work the authors described another allele of CESA5 different from the one used in our work and reported that this allele has no differences respect to WT. We added a sentence regarding this point.
Insufficient evidence is provided to conclude that the described genes (e.g. FLY1, CESA5, GL2) act as positive or negative regulators of seed development beyond what was previously published. Some inaccurate remarks are also present. Example: “mucilage is composed of a polar glycoprotein”. Most of Arabidopsis mucilage is pectin (specifically RG-I). No conclusive evidence is provided to support changes in RG-I branching (the sugar ratios are not enough).
- As the reviewer 2 suggests we have removed “mucilage is composed of a polar glycoprotein”, and we have changed it with a more accurate description; Please see lines: 404-407. Regarding the impact of these mutations on RG-I branching, these kinds of calculations are frequently used in literature to describe changes in pectin polymerization. In this case the ratios have calculated exclusively on pectin enriched fraction from mature seeds as reported in the result section 2.5. In the methods section 4.6 were described all the references for these calculations.
- The article is unusually long (18 pages before the methods) for the limited content of new results and the results/discussion text is too descriptive. For example, two pages of text describe the results of a single table (Table 1, section 2.7). The manuscript presents an unusually large amount of information that was previously known (example: Figure S3 and lines 421 to 455). In addition, most of the discussion reads like a broad literature review (e.g. section 3.1), rendering the overall scope of the manuscript ambiguous.
- We agree with the reviewer comment. We have reduced as possible the paragraph described the Table 1 and Figure S3. We have also reduced the overall manuscript, in particular the result and discussion sections.
We agree with the reviewer 2 regarding the information already revealed in literature and described in the Figure S3. However, this figure reports new data for luh stk, that was not previously described in literature. To facilitate reader in understanding, as well as to confirm previous information, we added those data supporting also the analysis of the double mutant.
It may be necessary to split the manuscript in two: One as a broader scope literature review, and another focused on gaining new insights into cell wall regulation during seed development (incorporating additional experiments and a more balanced interpretation).
- We have reduced the information described in the discussion. Moreover, the last paragraph 3.4. “Future perspectives in seed development” is reduced in the new version and it becomes the 5. “Conclusion” section of the manuscript as the guidelines of the journal suggest. We hope this new version satisfies the requirement to reduce the length of the manuscript.
- The data presentation must be improved. Fig. 1 and Fig. 2b graphs need separate Y-axes with their own values/units for the length/wide and area measurements. None of the figures in the supplementary information have legends. Table S1 has multiple references misspelled.
- We agree with this comment. We have used the same visual approach for the seed measurements, in past publication (please see: Di Marzo et al., 2022). We specified in the new version also in the legend of the Figure1 and 2 the values/units for the length/wide and area measurements. We have also added the legends in the Supplementary Figures file and corrected the misspelled references of the Table S1.
For Table 1, information about the replicates for the SD calculation is needed; “freshly harvested” should be defined (in terms of days or weeks post-harvest). The text in this figure is also too small.
- We agree with the reviewer comment. We have changed the text in the new version and add in the legend the number of replicates that we have used for the SD calculation. Moreover, we specify the differences between freshly and vernalized seeds in the Methods (see the section: 3. Germination test).
Minor points:
Line 243 – the three mentioned sugars would add up to more than 20-25% of total monosaccharides;
“Pectinic” is mentioned several times starting line 316 and should be changed to pectic
Line 328 – “confirms” is too strong; “suggests”
- We thank the reviewer 2 for the accurate revision. We have addressed all these minor points suggested by the reviewer.
Reviewer 3 Report
The article entitled "The genetic control of SEEDSTICK and LEUNIG-HOMOLOG in seed development: New insights into cell wall control" describes anatomical, physiological and biochemical studies on mutations in genes encoding for cell-wall modifying enzymes and selected transcription factors. The subject of the study is relevant and important for improving crop plants productivity. The indroduction is well written and interesting. Most of the data are collected and presented adequatly on the figures, methods are described in detail. The statistical analysis seems controversial in some cases and may lead to overinterpretation of the data. Fortunately the controversial parts may be, in my opinion, omitted or published as supplementary data.
I will briefly describe my considerations.
Major points:
1) the article is very long and so is the bibliography. I suggest making the story more concise.
2) the first two chapters of Results (lines 121-182 and Fig.1) are not convincing. The data were gathered and analyzed semi-automatically, as authors describe in Materials and Methods section, but still the differences shown are really small (but statistically significant). The reader may feel disappointed with the claims that the size of the seed is lower etc. if the difference is about 5% different from the control. What authors mean by biological replicate in measurements presented on Fig.1? Siliques/ seeds gathered from separate plants from one cultivation? or siliques / seeds gathered from several plants each time, cultivations repeated several times? Was the difference between the lines stable between the cultivations? Why only one allele was used for each gene (apart from fly)? In case of fly1.1 and fly1.3 the difference between the alleles is bigger than differences between control and other genes (and the reason is not well explained in the Discussion). In my opinion chapters 2.1 and 2.2 may be moved to supplementary materials and the discussion concerning these results (lines 474-575) shortened.
3) In line 203 in chapter 2.3 the influence of stk mutation on seed size is described but stk results are not shown on cited Fig.1, which makes the comparison difficult.
4) Later (lines 205-214) chapter 2.3 describes interesting data on seed coat structure of stk, luh and double mutant, but the data are (mostly) presented on supplementary figure S1 - why not in the main text?
5) the interesting phenotype of "inflated" seed of luh stk double mutant is shown on Fig. 2a. This effect is probably caused by some defects in cell wall rigidity, but it is not discussed in this chapter and is not discussed in Discussion section- why? Also the data concerning overall width of double mutant seeds presented on Fig. 2b suggest that the seed of luh stk is bigger than WT when hydrated, but more vulnerable to dessication.
6) chapter 2.4 describes interesting and important results on biochemical composition of cell walls in luh and stk mutant, I only propose to shorten first part (lines 236-268) to make the text more concise.
7) "the mutation is dominant over another mutation" (line 277, line 328)- I think should be "is epistatic",
8) Fig.3 and Material and Methods (lines 821-857) It is not well described how many samples were analyzed and how they were gathered, which stays in contrast to a very detailed description of the biochemical method.
9) Fig.4c, Fig. S2, Materials and methods (lines 816-820) and description of these results. The stk and luh mutations seem to control independent signaling pathways, still it is not described if or why not the Y2H test was performed only with AD fused to N-terminus of LUH and BD fused to N-terminus of STK. To completely exclude the protein interaction other combinations should be tested as well.
10) to make story more consised the initial paragraphs of chapter 2.6 (lines 360 to 395) may be moved to supplemental or radically shortened,
11) Fig. S3 shows interesting data on seed coat mucilage release. It is relevant to the main plot of the article- why it is shown in supplementary data and not in the main text?
Minor point:
1) Usually the Figures are not cited in discussion.
2) proposed model (Fig. 5) would be more clear if the hierarchy of the genes would be introduced (for example master regulators- transcription factors in separate line on the top, their interactors one line down etc.
Author Response
We want to kindly thank the reviewer 3 for the suggestions and comments.
The article entitled "The genetic control of SEEDSTICK and LEUNIG-HOMOLOG in seed development: New insights into cell wall control" describes anatomical, physiological and biochemical studies on mutations in genes encoding for cell-wall modifying enzymes and selected transcription factors. The subject of the study is relevant and important for improving crop plants productivity. The introduction is well written and interesting. Most of the data are collected and presented adequately on the figures, methods are described in detail. The statistical analysis seems controversial in some cases and may lead to overinterpretation of the data. Fortunately the controversial parts may be, in my opinion, omitted or published as supplementary data.
I will briefly describe my considerations.
Major points:
1) the article is very long and so is the bibliography. I suggest making the story more concise.
- We agree with the reviewer comment. We tried to reduce the amount of information described in the results and discussion sections. Moreover, the last paragraph 3.4. “Future perspectives in seed development” is reduced in the new version and it becomes the “Conclusion” section as the guidelines of the journal suggest. We hope that the story is more concise now in the new version of the manuscript.
2) the first two chapters of Results (lines 121-182 and Fig.1) are not convincing. The data were gathered and analyzed semi-automatically, as authors describe in Materials and Methods section, but still the differences shown are really small (but statistically significant). The reader may feel disappointed with the claims that the size of the seed is lower etc. if the difference is about 5% different from the control. What authors mean by biological replicate in measurements presented on Fig.1? Siliques/ seeds gathered from separate plants from one cultivation? or siliques / seeds gathered from several plants each time, cultivations repeated several times? Was the difference between the lines stable between the cultivations? Why only one allele was used for each gene (apart from fly)? In case of fly1.1 and fly1.3 the difference between the alleles is bigger than differences between control and other genes (and the reason is not well explained in the Discussion). In my opinion chapters 2.1 and 2.2 may be moved to supplementary materials and the discussion concerning these results (lines 474-575) shortened.
- We measured the seed and silique size with software which are frequently used by the scientific community to measure different phenotypes. We have used the Image J software in (Please see: Di Marzo et al., 2020, Cell Reports, Herrera-Ubaldo et al., 2019, Development) to test the silique length and in (Di Marzo et al., 2022, Journal of Experimental Botany) where with SmartGrain software, we also measured the seed size. ImageJ is frequently used for multiple phenotyping analyses, and it is well described the accuracy of this methos for plant phenotyping also in other species (Lino et al., 2008). SmartGrain is frequently used for seed shape measurements also in rice (example: Paul et al., 2020, Plant Direct). In the section 4.2 there were already the references for both software;
- We have performed another time the statistical analyses to make sure that also the small differences in size were statistically significant. The new statistical analyses confirm that there were differences in Area size and length for example for cesa5mutant seeds with respect to WT;
- We have specified in the text of the new versions the biological replicates and the pool of plants used for the collection of seed and siliques (Please, see section 4.2); This clarification has been inserted in the Fig1 and 2 legends of the revised version;
- We used one allele for each mutation (except for FLY1) because these alleles were already described in literature with a phenotype regarding mucilage capture formation and release;
- In the discussion section we have already reported the differences between the two alleles of FLY1 in mucilage release (Please see lines: 621-629);
- We agree that the manuscript was long. We reduced the chapters 2.1 and 2.2 in the revised version, but we prefer to leave them in the main text, since is necessary to make clear for the readers which are the phenotypes analyzed in the text, and which are the CW-related genes that have probably a role in seed shape and fruit length. From this first analysis (Fig.1) we decided to make the double mutant luh stk described in the Fig.2. Moreover, the pectin composition analyses (Fig.3 and Fig.4) were discussed considering the literature and the results described in Fig.1.
3) In line 203 in chapter 2.3 the influence of stk mutation on seed size is described but stk results are not shown on cited Fig.1, which makes the comparison difficult.
- We agree with the reviewer, however the stk measurements are already described in literature (Di Marzo et al., 2022) but also in the Fig. 2 where the readers can appreciate the measurements for this single mutant. In our opinion it would be repetitive also to insert them in the Fig.1.
4) Later (lines 205-214) chapter 2.3 describes interesting data on seed coat structure of stk, luh and double mutant, but the data are (mostly) presented on supplementary figure S1 - why not in the main text?
- We agree with the reviewer and we generated a new version of the Fig.2 with the detailed SEM pictures of the seed epidermal cells of the three mutant backgrounds with respect to WT.
5) the interesting phenotype of "inflated" seed of luh stk double mutant is shown on Fig. 2a. This effect is probably caused by some defects in cell wall rigidity, but it is not discussed in this chapter and is not discussed in Discussion section- why? Also the data concerning overall width of double mutant seeds presented on Fig. 2b suggest that the seed of luh stk is bigger than WT when hydrated, but more vulnerable to dessication.
- We thank the reviewer for the interesting point of view. In previously work, it was described the cell wall rigidity of stk seeds, we added this point in the new revised version, (Please see lines: 513,514). We added the consideration about the vulnerability to desiccation in the discussion section (Please see lines: 559-562);
6) chapter 2.4 describes interesting and important results on biochemical composition of cell walls in luh and stk mutant, I only propose to shorten first part (lines 236-268) to make the text more concise.
- We agree with the reviewer comment. We reduced the text of this chapter in the revised version.
7) "the mutation is dominant over another mutation" (line 277, line 328)- I think should be "is epistatic",
- We have followed the suggestion of the reviewer, and we changed it in the new version of the manuscript. (See line: 309,310).
8) Fig.3 and Material and Methods (lines 821-857) It is not well described how many samples were analyzed and how they were gathered, which stays in contrast to a very detailed description of the biochemical method.
- We added this information in the material and methods section 4.5 and in the Figure 3 and 4 legends.
9) Fig.4c, Fig. S2, Materials and methods (lines 816-820) and description of these results. The stk and luh mutations seem to control independent signaling pathways, still it is not described if or why not the Y2H test was performed only with AD fused to N-terminus of LUH and BD fused to N-terminus of STK. To completely exclude the protein interaction other combinations should be tested as well.
- The Y2H experiment was tested using GAL4 AD fused to the N-terminus region of LUH or STK, and GAL 4 BD fused to the N-terminus of SEP3. We added this information in the methods section. Interactions are usually tested in one way. The YH2 system used here minimizes the effect of AD- or BD-fusion orientation on protein interactions.
10) to make story more consised the initial paragraphs of chapter 2.6 (lines 360 to 395) may be moved to supplemental or radically shortened,
- We made more concise the initial part of the chapter 2.6 as requested by the reviewer 3 and in general all the chapter 2.6.
11) Fig. S3 shows interesting data on seed coat mucilage release. It is relevant to the main plot of the article- why it is shown in supplementary data and not in the main text?
- We decided to put these data in the Fig. S3 because some of the phenotypes are already described in literature and the new data provided here is that luh stk reassembles the phenotype of the single stk
Minor point:
1) Usually the Figures are not cited in discussion.
- We propose to keep these specifications here to help the readers in finding the data in the text.
2) proposed model (Fig. 5) would be more clear if the hierarchy of the genes would be introduced (for example master regulators- transcription factors in separate line on the top, their interactors one line down etc.
- We changed the model to make more clear the differences between transcriptional regulators and enzymes in the Fig.5 as the reviewer 3 suggests.
Reviewer 4 Report
See enclosure.

Author Response
We want to kindly thank the reviewer 4 for the suggestions and comments.
This manuscript reports on the role of two transcriptional regulators, SEEDSTICK and LEUNIG-HOMOLOG), in seed development. It aims at showing genetics evidence for a link between them and cell wall players involved in polysaccharide biosynthesis or modification, homogalacturonan, cellulose and xyloglucan.
Major comments
The title does not really fit with the content of the article. The article also deals with the development of the fruit. Then, the meaning of seed development needs to be defined: is germination included? What means “cell wall control”?
- We agree with the reviewer comment, and we decided to add in the title of the manuscript also the fruit development. To better define the meaning of seed development, in the new Conclusion section, the Fig. 5 has as title “Seed growth and germination”. “Cell wall control” means the possible impact of different genes involved in cell wall regulation on fruit, seed growth and germination. We don’t have an alternative manner to make short this idea.
In the introduction, two different processes are mixed (lines 55-58), the development of the seed including that of the embryo from the globular to the green mature stage, and the germination starting with the rupture of the testa and that of the endosperm. This should be clarified.
- We agree with the reviewer. We have introduced a link in the revised version to clarify this (Please see lines: 49,50).
To clarify the objectives of work and to avoid redundancies, I suggest the authors to include a scheme to show the cascades of regulations and the possible roles of each gene/protein studied already in the introduction (see for example ref. 31, Figure 3; or ref. 37, Figure 3). FLY1 should also be mentioned in the introduction (line 145). The last paragraph of the introduction does not clearly explain the objectives of the work.
- We agree with the reviewer. The main text is already long and in the Fig.5 model the readers can appreciate in part the cascades of regulation and the specific role of each gene.
- We mentioned which protein is encoded from FLY1 gene in the introduction, (see line: 100);
- As the reviewer 4 suggests we changed the last paragraph of the introduction (Please, see lines: 118-122).
I have difficulties to see how general conclusions can be drawn regarding the role of cell walls in seed development because all the seed tissues (embyos and teguments) are mixed for the analyses of the cell wall composition. A detailed analysis of the different tissues with antibodies raised against specific cell wall epitopes would help understanding what is observed in the mutants. Seed development cannot be reduced to the scheme presented in Figure 5.
- We want to thank the reviewer 4 for the interesting suggestion. The outcome of the antibody analyses will likely be inconclusive unless done on many different mutants, stages (seed growth and germination) and with a wide range of antibodies (which in specific target polymers are more or less effective in those tissues). For that reason and considering the main goal of this manuscript we reasoned that is not feasible at the current status.
As far as I know, PRX36 is not involved in pectin metabolism. Please check this information (line 97 and at other places in the manuscript).
- According to the reviewer suggestion, we corrected this point in the new version of the manuscript. (See for example lines: 101-104).
I also have difficulties to figure out the role of PRX36 and PMEI6 in the growth of the seed. Indeed, these genes have been shown to be specifically expressed in the mucilage mother cells at a late stage of seed development, and to play roles in mucilage release (Kunieda et al. 2013, Plant Cell 25: 1355; Francoz et al. 2019, Dev Cell 48: 261).
- We thank the reviewer for this point. STK, PMEI6 and PER36 are expressed in the epidermal outer layer of the seed-coat and influence the final seed size. In fact, the three single mutants are smaller with respect to WT. STK, for example is expressed in seed outer integuments upon fertilization till 2 DAP and remains active in the epidermal outer layer of the seed coat up to 6 DAP (Ezquer et al., 2016). The PMEI6 is expressed specifically in the epidermal cells of the seed coat where mucilage polysaccharides accumulate (Maximum expression from 8 to 10 DAP) and remains active until 14 DAP (Saez-Aguayo et al., 2013). PER36 begins to accumulate at 5 DAP and reaches the maximum level at 6-7 DAP (Kunieda et al., 2013).
It has been previously shown that prx36 exhibits defects in mucilage release (Kunieda et al. 2013; Francoz et al. 2019). However, this is not clear from what is observed in Figure S3 (line 448).
- We thank the reviewer for this point. The methodology used by Kunieda et al., 2013 and in our work are substantially different. Kunieda and colleagues applied 2h of shaking previously to observation, while in our work we performed the staining without this step. Nevertheless, Kunieda already reported variability of this allele in terms of mucilage release.
Seed germination (Table 1): how many independent biological replicates were considered and were the seeds obtained at different seasons? It is well known that the conditions of seed production strongly affect their capacity to germinate. Besides, what was observed? Was it the rupture of testa or that of endosperm?
- In the new version, we added the info. about the biological replicates in the legend of the figures, but this is also described in the Material section. We considered for germination test, as already described in the first version of the manuscript in the section 4.3, the testa rupture. The Arabidopsis plants are growing in chambers with controlled conditions. Following the OGM rules we are not allowed to growth these plants in open field “seasonality”.
More generally, the number and the type of biological replicates should be more precisely indicated in Material and Methods and in legends to the figures. For example, 10 biological replicates are mentioned in legend to Figure 2d and 3 biological replicates of 10 siliques each in Material and Methods.
- We have added as the reviewer suggests all these informations in the material section but also in the Figure and Table legends of each experiment.
A pectin-enriched fraction has been obtained instead of a “pure pectin fraction” (line 318).
- To make shorter this paragraph we eliminated this part. At the beginning of the section there is pectin-enriched fraction as the reviewer suggests.
The presence of some polysaccharides can be inferred from that of specific monosaccharides (line 312). For example, Rha is specific for rhamnogalacturonans. There are some formulae which allow giving some characteristics of the polysaccharides (see Houben et al. 2011, Carbohydr Res 346: 1105). It would be interesting to make these calculations.
- We thank the reviewer for this suggestion. We added in the new version of the manuscript the calculation of the pectin linearity ratio GalA/FRAGX. Please, see lines 318-325 and Figure 4b.
As mentioned at several places in this report, the literature content should be better taken into account in the whole manuscript.
Due to my difficulties in understanding part of the results, I must say that I did not read the discussion in details. I only want to mention that it should be better focused on the objective of the manuscript rather on general considerations on the role of the cell wall in developmental processes. There are also some shortcuts like lines 533-535 where it is mentioned that PRX36 may play a role in the methylesterification of homogalacturonans. To propose a model integrating all the results, it would be important to know in which tissues all the studied genes are expressed and at which stage of seed development. The other point is to define seed development. I think that it cannot be limited to seed growth.
- We thank the reviewer for these suggestions. We edited the discussion section, and we pinpointed on the objective of the manuscript. We have removed in the revised version also the shortcut of PER36. Moreover, all the genes described in this manuscript are expressed exclusively or not in the seed-coat, as previously described. Finally, we better describe in the introduction but also in the Model, the differences between seed growth and germination.
Additional comments
- The lettering in the figures is too small. Some panels are also too small (Figure 1, panel b; Figure 2, panel c; Figure 4, panel c).
- We thank the reviewer. We have modified the Figures according to the reviewer 4 suggestions.
- The lettering of Table 1 is too small. Is it reasonable to mention two digits after the comma regarding the precision of the measurements? Please add legends to figures in the supplementary data file. Put the gene names in italics.
- We agree with the reviewer 4. We edited the Table 1 and removed the two digits. We have also added the legends in the Supplementary data file.
- Please check the references: all Latin names should be in italics. Titles should be decapitalized, in particular the plant species names should not begin with a capital letter (e.g. refs 12, 26, 39, ...). Gene names should be in italics (e.g. refs 11, 76, 96, 102, 103, 111). Refs 43, 45, 58, 72, 80 and 113 are incomplete. Check journal names (e.g. Proc Natl Acad Sci USA, Plant Cell, Environ). Ref 43: Plant Mol Biol 47: 9-27. Ref 30: North, H.M. Ref96: BSISTER.
- We checked all the references, and we edited them in the new version of the manuscript.
- What means “pectinic compound”?
- We changed it to “pectic compound”, (line: 399).
- What is the polar glycoprotein present in the mucilage? Ref; 64 does not mention this glycoprotein.
- We thank the reviewer for this point. We better described this point in the new version of the manuscript (see lines: 399-409).
- Use the present reference names for galacturonic acid (GalA instead of GalU) and glucuronic acid (GlcA instead of GlcU) everywhere (see https://www.ncbi.nlm.nih.gov/glycans/snfg.html).
- We changed it according to reviewer 4 suggestion.
- “h” and ‘min” instead of “hours” and “minutes” everywhere.
- We changed it according to reviewer 4 suggestion.
- Add city and country for all the purchasers in Material and methods.
- We added in detail the info. regarding the reagents that we have used, and in some cases also the websites where the readers can find the product. Moreover, in the Table S1 we added the cities and countries for all the Group that send us seed stocks.
- Please, indicate in all the legends to the figures whether what is shown is representative of the observations.
- We added it according to reviewer 4 suggestion.
- lines 40, 826: mass instead of weight.
- We changed it according to reviewer 4 suggestion.
- line 62: Please rephrase, cellulose is not synthesized in the plasma membrane.
- We rephrased it. (Please see lines: 64-66)
- line 70: xylogalacturonan
- We changed it according to reviewer 4 suggestion.
- line 75: Add ref. 28.
- We added this reference according to reviewer 4 suggestion.
- line 95: define FEI (see line 130).
- We defined it in the line 98 of the introduction section.
- line 99: encodes the only...
- We have removed this according to reviewer 4 suggestion.
- line 106: Ref. 43 does not discuss the relationships between seed and fruit development.
- We have removed it according to reviewer 4 suggestion.
- line 113: Please rephrase, what means “molecular regulation of the cell wall”?
- We edited, with a new paragraph, the last part of the introduction in the new version.
- line 114: Explain what are “TF-CW interactions”.
- We removed “molecular regulation of the cell wall” and explain better the scope of our manuscript in the last paragraph of the introduction. (Please, see lines: 118-122).
- line 125: Can cell walls be considered as a network?
- We changed it with “complex structure” (Please see line: 127)
- line 146: Information is missing. The sentence should include “by recycling unprocessed PME enzymes in the endomembrane system of the seed coat epidermis” (see ref. 52).
- We added this sentence as the reviewer 4 suggests in the lines 147,148.
- line 150: “mutation in the class III peroxidase PER36 gene (per36) leads...” - line 160: remove (beta-xylosidase1).
- We have removed it as the reviewer 4 suggests.
- line 181: Please define “CW regulatory factors”.
- We removed it.
- line 201: Please rephrase “seed coat mucilage demethylesterification”.
- We removed it.
- line 207: “irregular shapes”
- We changed it as the reviewer 4 suggests.
- line 211: Which frequency?
- We thank the reviewer for the suggestion. We removed “frequency” and we added a more precise description of the phenotype. (Please see lines: 203-214).
- line 213: What means “intermediate phenotype”?
- We describe an intermediate phenotype when, for example for the length of the double mutant seeds, these seeds are longer than stk but shorter with respect to luh. We specified it in the lines 231-233.
-line 309: “proteins” instead of “protein”
- We changed it as the reviewer 4 suggests.
- line 310: “polymers” instead of “elements”?
- We changed it as the reviewer 4 suggests.
- line 832: Why “successively”? - We removed “successively”;
- line 833: 4000 g - We changed it.
- line 835: explain E.S. - We better explain this point in the new version of the manuscript. (Please, see lines: 709-711).
Reviewer 5 Report
in Discussion part, the authors could be introduce its upstream regulation of non-coding RNA for these TFs, especially in stk and luh. it will be present more mechniasm for these TFs regulating seed development.
Author Response
in Discussion part, the authors could be introduce its upstream regulation of non-coding RNA for these TFs, especially in stk and luh. it will be present more mechniasm for these TFs regulating seed development.
- We want to kindly thank the reviewer 5 for this suggestion. However, the focus of the manuscript is not the upstream regulation of these transcriptional regulators by non-coding RNA. It would be interesting to add this point to the discussion section, but the other reviewers have requested making shorter the discussion section and in general the manuscript.
Round 2
Reviewer 2 Report
The authors have significantly improved the main and supplemental figures, adding important clarifications in the legends.
Most of my comments were adequately addressed for publication. The first part of my first major comment may have been misunderstood. I did not doubt the accuracy of the statistical tests but wanted to stress the importance of describing the magnitude of the size differences in the text. A 5% difference, even if significantly different, would indicate a relatively small effect.
Some sections of text, such as in the Discussion, are still too long and readers would benefit from a more concise version.
"Vernalized" should be rephrased since it applies to plants prior to flowering, not to seed storage.
Reviewer 3 Report
I read the improved version of the manuscript entitled : "The genetic control of SEEDSTICK and LEUNIG-HOMOLOG in seed and fruit development: New insights into cell wall control". I would like to thank the authors for introducing changes in the text, in particular claryfying some methodological considerations in Material and Methods section, in Figure captions and in the model (Fig.5).
Unfortunately, I do not feel that all my considerations were adressed in the improved text or reviewer's response letter. Major points are:
1) The authors stated how many biological replicates and how many independent plant cultivations were performed for obtaining data for seed germination analysis and sugar content analysis (2 to 4 independent biological replicates, which I understand as independent cultivations - am I right?) and this number is satisfactory to gain unbiased results if the differences between genotypes is large and the variance inside the genotype is small - and so they are. Thank for this details. Still the data presented on Fig. 1 and Fig.2 come from ONE biological replicate and the difference between genotypes is very small. I understand the measurement by already published software is perfectly OK and so is the statistical method applied. Only the design of the whole experiment is not proper. The authors have not answered how many cultivations were performed and have not commented what was the variance between the results obtained from different cultivations for a given genotype. "Representative result" presentation is usually used for non-numerary data such as Western blot or microscopic image. May the authors straightforwardly state the number of cultivations and comment on the variation between the cultivations for one genotype?
2) I still consider the answer to my question about Y2H unsatisfactory. Fusions of proteins/ protein domains to one's protein of interest, which is used in such methods as Y2H, bi-fluorescent complementation or FRET is always tricky. If positive and negative controls work (as in author's case) and there IS an interaction- than the experiment is informative. If controls work, but there IS NO interaction- it does not mean there is no real interaction. The real interaction can only be prevented by introducing fusion proteins / protein domains and masking the site of interaction of the partners. In such case the whole system must be recloned to get the additional domains on the other terminus of the proteins or exchange the bait and prey. In the case presented in the article authors should consider repeating the Y2H experiment after recloning or remove the image and conclusions. The fact that other protein (Sepallata 3) interacts with one of the partners (Seedstick) does not mean that the system is fine to study interaction with another protein partner (Leunig homologue), bacause the site of interaction may be different on protein (Seedstick) surface.